# Topological Defect Lines in Two-Dimensional Fermionic CFTs

Chi-Ming Chang,[a,b] Jin Chen,[c,a] and Fengjun Xu[a,b]

[a] *Yau Mathematical Sciences Center (YMSC), Tsinghua University, Beijing, 100084, China*

[b] *Beijing Institute of Mathematical Sciences and Applications (BIMSA), Beijing, 101408, China*

[c] *Department of Physics, Xiamen University, Xiamen, 361005, China*

cmchang@tsinghua.edu.cn, zenofox@gmail.com, xufengjun321@gmail.com

## Abstract

We consider topological defect lines (TDLs) in two-dimensional fermionic conformal field theories (CFTs). Besides inheriting all the properties of TDLs in bosonic CFTs, TDLs in fermionic CFTs could host fermionic defect operators at their endpoints and junctions. Furthermore, there is a new type of TDLs, called q-type TDLs, that have no analog in bosonic CFTs. Their distinguishing feature is an extra one-dimensional Majorana fermion living on the worldline of the TDLs. The properties of TDLs in fermionic CFTs are captured in the mathematical language of the super fusion category. We propose a classification of the rank-2 super fusion categories generalizing the $\mathbb{Z}_8$ classification for the anomalies of $\mathbb{Z}_2$ symmetry. We explicitly solve the F-moves for all the nontrivial categories, and derive the corresponding spin selection rules that constrain the spectrum of the defect operators. We find the full set of TDLs in the standard fermionic minimal models and a partial set of TDLs in the two exceptional models, which give CFT realizations to the rank-2 super fusion categories. Finally, we discuss a constraint on the renormalization group flow that preserves a q-type TDL.

# 1   Introduction

It goes without saying that symmetry is a central theme in theoretical physics as it provides strong constraints on the dynamics of physical systems. A textbook example is Noether's theorem in classical physics, which states that every continuous symmetry is associated with a conservation law. At the quantum level, 't Hooft anomalies of internal or spacetime symmetries give non-trivial constraints on the infrared (IR) fixed points of renormalization group (RG) flows, especially when the IR theories are strongly-coupled.

Recently, the understanding of symmetry has significantly evolved. In the modern point of view, the presence of symmetries in a quantum field theory (QFT) is equivalent to the existence of topological defects [1]. More precisely, given a global symmetry $G$ in a $D$-dimensional QFT, the action of a group element $g \in G$ on a local operator $\mathcal{O}$ is implemented by shrinking a topological defect $U_g$ inserted on a $(D-1)$-sphere with the operator $\mathcal{O}$ placed at a point lying inside the sphere, as shown in the figure below[1]

$$
\begin{array}{ccc}
\bullet\,\mathcal{O} & = & \bullet\,\widehat{U}_g(\mathcal{O}) \\
U_g & &
\end{array}
\tag{1.1}
$$

The defect $U_g$ is called a topological defect because physical observables are invariant under continuous deformations of the manifold where $U_g$ is inserted on. The group multiplication of two elements $g_1$, $g_2 \in G$ is realized by fusing the corresponding topological defects $U_{g_1}$ and $U_{g_2}$.

With such a conceptual breakthrough, the notion of symmetry has been extended in various directions including higher-form symmetry [1], higher-group symmetry [2–6], and category symmetry (non-invertible symmetry) [7, 8], where the last one is the main subject of this paper. A category symmetry generalizes the group structure of an ordinary symmetry by including topological defects that do not have an inverse under the fusion. The prototype example is the symmetry generated by the topological defect lines (TDLs) in the two-dimensional Ising conformal field theory (CFT), where the TDL implementing the Kramers-Wannier duality is known to be non-invertible [8–11]. The generalization from ordinary to category symmetry is expected to provide us with new tools to understand and constrain various physical systems. For instance, the generalization of 't Hooft anomaly matching condition for category symmetry had been used to constrain the RG flows between CFTs or gapped phases [8, 12–17]. Non-invertible defects also play important roles in the study of quantum gravity [18–23]. In two dimensions, category symmetries (topological defect lines) are ubiquitous, and there is a comprehensive study of them [7–12, 14, 24–54]. Viewing symmetries as TDLs allows one to discuss junctions and endpoints of the TDLs, which are described in the mathematical language of the fusion category. In certain cases, this 2d picture can be lifted to the framework of 3d TQFTs. More precisely, in the boundary-bulk correspondence between 2d RCFTs (rational CFTs) and 3d TQFTs [55, 56], the Verlinde lines (TDLs preserving the maximal chiral algebra) can be naturally interpreted as anyons in three dimensions. Recent developments further extend the study of category symmetry to higher dimensional QFTs [57–67].

---

[1]We consider zero-form symmetries throughout this paper.

In this work, we extend the study of category symmetry in a different direction to two-dimensional fermionic CFTs. In general, fermionic CFTs have a richer structure than bosonic CFTs due to the existence of fermionic local operators, which are Grassmann-valued operators with half-integer spins. On Riemann surfaces, the fermionic local operators obey anti-periodic (Neveu-Schwarz) or periodic (Ramond) boundary conditions along non-contractible cycles. These structures carry over to the defect operators living at the junctions or endpoints of the TDLs, except that fermionic defect operators could in general have non-half-integer spins. A universal example in all fermionic CFTs is the TDL associated with the fermion parity $\mathbb{Z}_2$ symmetry, which assigns $+1$ (or $-1$) charge to bosonic (or fermionic) local operators. The defect operators living at the endpoints of it are nothing but the Ramond sector operators.[2]

The existence of fermionic defect operators leads to several modifications and additions to the properties of TDLs. In particular, they can live at trivalent junctions and be involved in the F-moves (the crossings) of the TDLs. Due to their fermion statistics, one needs to introduce an ordering to the junctions to keep track of the ordering of the fermionic defect operators in the correlation functions. Exchanging the order would give extra signs, schematically as

$$\psi \underset{1}{\succ}\!\!\!-\!\!\!\prec_{2} \psi' = \langle \cdots \psi \psi' \cdots \rangle = -\langle \cdots \psi' \psi \cdots \rangle = - \, \psi \underset{2}{\succ}\!\!\!-\!\!\!\prec_{1} \psi' \ . \tag{1.2}$$

As a result, the F-moves now satisfy a different consistency condition, called the super pentagon identity, instead of the pentagon identity for the TDLs in bosonic CFTs. Besides the super pentagon identity, the F-moves are also constrained by a new projection condition if the internal lines of the H-junctions belong to a new type of TDLs, called q-type.[3] Q-type TDLs can host a one-dimensional Majorana fermion on their worldlines, which can be pair-created or pair-annihilated as

$$\Big| \ = \ \begin{matrix} {}_1 \bullet \ \psi \\ {}_2 \bullet \ \psi \end{matrix} \ . \tag{1.3}$$

The aforementioned projection condition follows from the pair-creation on the q-type internal TDLs.

---

[2]In the example of fermionic minimal models [68, 69], one can easily check that the spin and statistics of the Ramond sector operators are uncorrelated.

[3]The q-type object in super fusion category was introduced in [70, 71] and further studied in [72] in the study of 2+1d fermionic topological phases.

The existence of q-type TDLs leads to many interesting consequences. For instance, the notion of invertibility for q-type TDLs is slightly different from the one for the m-type TDLs. Due to the 1d Majorana fermion $\psi$, the defect Hilbert space of a q-type TDL $\mathcal{L}_q$ has doubly degenerate ground states, one bosonic and one fermionic. As a result, the fusion of $\mathcal{L}_q$ with its orientation reversal takes the form

$$\mathcal{L}_q \overline{\mathcal{L}}_q = 2I + \cdots , \tag{1.4}$$

where the $\cdots$ denotes some nontrivial TDLs. We can see that, even when the $\cdots$ is absent, the fusion rule (1.4) does not take the form of a group multiplication law due to the factor of 2 in front of the trivial TDL $I$. Nevertheless, as we will see in Section 2.2.6, one could recover the group multiplication by a proper rescaling of the q-type TDL $\mathcal{L}_q$. Hence, we could define the invertible q-type TDL as the q-type TDL satisfying the fusion rule (1.4) with $\cdots$ absent. Due to the rescaling, the invertible q-type TDLs satisfy a modified Cardy condition, which was used in finding invertible q-type TDLs in many fermionic CFTs [16, 17].

Non-invertible q-type TDLs are also ubiquitous in fermionic CFTs. To study them, in Section 3, we focus on systems of TDLs involving only one nontrivial simple m-type or q-type TDL, which generates a $\mathbb{Z}_2$ (invertible) symmetry or a rank-2 non-invertible symmetry with a Fibonacci-like fusion rule. By solving the super pentagon identity and the projection condition for low multiplicity, for the $\mathbb{Z}_2$ symmetry case, we find eight solutions of the F-moves, which reproduce the $\mathbb{Z}_8$ classification of the anomalies of $\mathbb{Z}_2$ symmetry [73–77]. For the rank-2 non-invertible symmetry case, we find ten solutions of the F-moves, which contain all the F-moves obtained from fermion condensation in [78]. We further conjecture that these 18 solutions give all the rank-2 super fusion categories (summarized in Table 1 and 2). One important application of explicit solutions of the F-moves is that they could be used to constrain the spin spectrum of the defect Hilbert space [8]. Following [8], we derive spin selection rules for the nontrivial TDLs (see Section 4), which give us a handle to identify TDLs in CFT realizations. The simplest family of fermionic CFTs are the standard and exceptional fermionic minimal models [68, 69, 79]. We found the full set of simple TDLs in the standard models and a partial set of TDLs in the exceptional models (see Section 5). In particular, there are q-type TDLs in all the standard models with $m = 0, 3 \mod 4$, and all the exceptional models. The fermionic minimal models provide realizations of many non-invertible rank-2 super fusion categories (see Table 2).

The mathematical structure of the fermionic defect operators and the q-type TDLs are captured in the language of the super fusion category [80, 81]. It is also instructive to relate TDLs in 2d fermionic RCFTs to anyons in 3d spin TQFTs, which can be obtained by gauging a mixture of a $\mathbb{Z}_2$ one-form symmetry and the spin structure in certain bosonic TQFTs, or in the language of condensed matter, a fermionic anyon condensation [70]. In this picture, the q-type objects are fixed points under the action of the $\mathbb{Z}_2$ one-form symmetry.

This paper is organized as follows. Section 1.1 briefly reviews the defining properties of TDLs in 2d bosonic CFTs. Section 2.1 introduces the defining properties of TDLs in 2d fermionic CFTs including the fermionic defect operator and the q-type TDL. Section 2.2 derives several consequences from the defining properties including the F-moves, the super pentagon identity, the universal sector of the F-moves, and the modified Cardy condition for q-type invertible TDLs. Section 2.3 relates the TDLs in 2d fermionic CFTs to the super fusion categories. Section 3 solves the super pentagon identity for the rank-2 super fusion categories up to multiplicity 2, and conjectures a classification of the rank-2 super fusion categories. Section 4 derives the spin selection rules for the spectrum of defect operators in all the non-trivial rank-2 super fusion categories. Section 5 gives the full set of simple TDLs in the standard fermionic minimal models and a partial set of TDLs in the exceptional models, and discusses how they realize the rank-2 super fusion categories. Section 6 ends with a summary and comments on the constraints of the fermionic RG flows from the TDLs. Appendix A gives a solution to the projection condition. Appendix B presents the solution to the universal sector of oriented q-type TDLs. Appendix C gives a detailed derivation of the formula of the fusion coefficients in Section 2.2.5. Appendix D gives the detailed data of the TDLs in the super fusion category $\mathcal{C}_{\mathrm{q}}^2$. Appendix E gives the F-moves of the super fusion category that shows up in the $m = 4$ fermionic minimal CFT, which is of rank-4 and can be regarded as the "tensor product" of two rank-2 super fusion categories discussed in Section 3.

## 1.1 Review of TDLs in 2d bosonic CFTs

In a two-dimensional CFT, a global symmetry $G$ can be implemented by topological defect lines (TDLs) [7–10,25–42], supporting on oriented paths. When a charged operator $\mathcal{O}$ passes through a TDL $\mathcal{L}_g$ associated with a group element $g \in G$, it gets acted by a $g$-transformation, denoted by $\widehat{\mathcal{L}}_g(\mathcal{O})$. Graphically, we have

$$
\begin{array}{ccc}
\mathcal{L}_g & & \mathcal{L}_g \\
\Big\uparrow \quad \overset{\bullet}{\mathcal{O}} & = & \overset{\bullet}{\widehat{\mathcal{L}}_g(\mathcal{O})} \quad \Big\uparrow
\end{array}
\tag{1.5}
$$

The group multiplication of $G$ is realized by the fusion of the TDLs,

$$\mathcal{L}_g \quad \mathcal{L}_{g'} \qquad \mathcal{L}_{gg'}$$

$$\Big| \quad \Big| \quad = \quad \Big| \tag{1.6}$$

where we simply bring the two TDLs on top of each other. The fusion satisfies all the group theory axioms: it is associative, the identity element is realized by the trivial TDL $I$, and finally the TDL $\mathcal{L}_{g^{-1}}$ associated with the inverse of $g$ is the orientation reversal $\overline{\mathcal{L}}_g$ of the TDL $\mathcal{L}_g$.

In the language of TDLs, it is natural to generalise the group structure of $G$ to a (semi)ring structure, which comprises a direct sum $+$ in addition to the fusion product. A general fusion takes the form

$$\mathcal{L}_a \mathcal{L}_b = \sum_c N_{ab}^c \mathcal{L}_c \,, \tag{1.7}$$

where the RHS is a sum of simple TDLs $\mathcal{L}_c$, which cannot be decomposed further, and the fusion coefficients $N_{ab}^c$ are nonnegative integers. In general, a TDL $\mathcal{L}_a$ could be non-invertible under the fusion with, for example, the fusion rule

$$\mathcal{L}_a \overline{\mathcal{L}}_a = I + \mathcal{L}_b + \cdots \,. \tag{1.8}$$

When a charged operator $\mathcal{O}$ passes through the non-invertible TDL $\mathcal{L}_a$, by applying the fusion rule (1.8) piece-wisely, one finds

$$\mathcal{L}_a \qquad\qquad \mathcal{L}_a \qquad\qquad \mathcal{L}_a$$

$$\Big| \bullet \mathcal{O} \quad \propto \quad \underset{\widehat{\mathcal{L}}_a(\mathcal{O})}{\bullet\Big|} \quad + \quad \underset{\widehat{\mathcal{L}}_a^v(\mathcal{O})}{\bullet\!\xrightarrow{\mathcal{L}_b}\Big|} v \quad + \quad \cdots \,. \tag{1.9}$$

This makes us inevitably discuss junctions and endpoints of TDLs, whose properties will be reviewed in the latter part of this section. Here, let us briefly explain the various ingredients of (1.9). The trivalent junction of the outgoing TDLs $\mathcal{L}_a$, $\overline{\mathcal{L}}_a$, $\overline{\mathcal{L}}_b$ and the endpoint (or one-way junction) of the TDL $\mathcal{L}_b$ host defect Hilbert spaces $\mathcal{H}_{\mathcal{L}_a,\overline{\mathcal{L}}_a,\overline{\mathcal{L}}_b}$ and $\mathcal{H}_{\mathcal{L}_b}$, respectively. $v$ is a topological (zero conformal weight scalar) defect operator in the defect Hilbert space $\mathcal{H}_{\mathcal{L}_a,\overline{\mathcal{L}}_a,\overline{\mathcal{L}}_b}$. The TDL $\mathcal{L}_a$ induces a linear map $\widehat{\mathcal{L}}_a^v$ from the Hilbert space $\mathcal{H}$ of local operators to the defect Hilbert space $\mathcal{H}_{\mathcal{L}_b}$, i.e. $\widehat{\mathcal{L}}_a^v(\mathcal{O}) \in \mathcal{H}_{\mathcal{L}_b}$. The sequence of maps $(\widehat{\mathcal{L}}_a, \widehat{\mathcal{L}}_a^v, \cdots)$ characterize the category symmetry action of the non-invertible TDL $\mathcal{L}_a$ on local operators.

To be self-contained, we give an overview of the defining properties of TDLs in [8] that would be relevant for our discussions in the following sections.

- **Isotopy invariance:** The deformations of TDLs by the ambient isotopies of the graph embedding on the flat spacetime do not change physical observables such as correlation functions involving these TDLs.[4] It leads to the fact that TDLs commute with the stress-tensor.

- **Defect operators/states:** TDLs can join at a point-like junction, which is equipped with a defect Hilbert space, e.g. $\mathcal{H}_{\mathcal{L}_1, \mathcal{L}_2, \cdots, \mathcal{L}_k}$ for the $k$-way junction of the TDLs $\mathcal{L}_1, \mathcal{L}_2, \cdots, \mathcal{L}_k$.[5] The defect Hilbert space inherits the addition from the associated TDLs, for example $\mathcal{H}_{\mathcal{L}_1 + \mathcal{L}_2} = \mathcal{H}_{\mathcal{L}_1} \oplus \mathcal{H}_{\mathcal{L}_2}$. Since TDLs commute with the stress-tensor, the defect Hilbert space forms a representation of the left and right Virasoro algebras. By the state/operator map, the states in the defect Hilbert space correspond to point-like defect operators living at the junction of the TDLs. The junction vector space $V_{\mathcal{L}_1, \mathcal{L}_2, \cdots, \mathcal{L}_k}$ is defined to be the conformal weights $(h, \tilde{h}) = (0, 0)$ subspace of $\mathcal{H}_{\mathcal{L}_1, \mathcal{L}_2, \cdots, \mathcal{L}_k}$. In particular, the junction vector $m \in V_{\mathcal{L}_1, \mathcal{L}_2}$ at the two-way junction of the TDLs $\mathcal{L}_1$ and $\mathcal{L}_2$ gives a linear map[6]

$$m : H_{\mathcal{L}_1} \to H_{\mathcal{L}_2} . \tag{1.10}$$

Note that each junction comes with an ordering of the TDLs attached to the junction. Following [8], we mark the last TDL entering each junction by a cross $\times$.

- **Locality:** A TDL configuration on a Riemann surface is invariant under the cutting and gluing. One can cut the Riemann surface along a circle that intersects the TDLs transversely, and put on the circle a complete orthonormal basis of states in the Hilbert space associated with that circle. The states on the circle can be constructed by taking a disc with (defect) operators inserted, which could be further glued back to the Riemann surface with a circle boundary. For example, consider a TDL configuration that contains an H-junction on a Riemann sphere $S^2$, as shown in (1.11). Two junction vectors $v_1 \in V_{\mathcal{L}_1, \mathcal{L}_2, \overline{\mathcal{L}}_5}$ and $v_2 \in V_{\mathcal{L}_3, \mathcal{L}_4, \overline{\mathcal{L}}_5}$ are inserted at the two trivalent junctions of the H-junction. One can cut off a disc containing the H-junction and glue back a disc

---

[4]Note that on curved spacetime, the deformations of TDLs might lead to phases, called isotopy anomalies [8].

[5]The cyclic permutation of $\mathcal{L}_1, \mathcal{L}_2, \cdots, \mathcal{L}_k$ defines an isomorphic between the defect Hilbert space $\mathcal{H}_{\mathcal{L}_1, \mathcal{L}_2, \cdots, \mathcal{L}_k}$.

[6]In terms of fusion category, these operators define morphisms in $\text{Hom}(\mathcal{L}_1, \mathcal{L}_2)$.

with a four-way junction inserted inside, graphically duplicated as

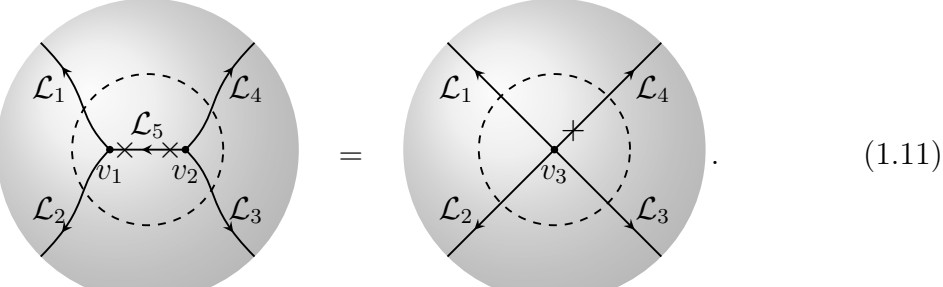

$$(1.11)$$

The junction vector $v_3 \in V_{\mathcal{L}_1,\mathcal{L}_2,\mathcal{L}_3,\mathcal{L}_4}$ is determined by the junction vectors $v_1$ and $v_2$ via a bilinear map $H_{\mathcal{L}_2,\mathcal{L}_3}^{\mathcal{L}_1,\mathcal{L}_4}(\mathcal{L}_5) : V_{\mathcal{L}_1,\mathcal{L}_2,\overline{\mathcal{L}}_5} \otimes V_{\mathcal{L}_3,\mathcal{L}_4,\mathcal{L}_5} \to V_{\mathcal{L}_1,\mathcal{L}_2,\mathcal{L}_3,\mathcal{L}_4}$, which is given by the disc "correlation functional":

$$H_{\mathcal{L}_2,\mathcal{L}_3}^{\mathcal{L}_1,\mathcal{L}_4}(\mathcal{L}_5) \equiv \quad\quad\quad\quad\quad\quad\quad\quad\quad . \quad\quad (1.12)$$

It takes the input of two junction vectors in $V_{\mathcal{L}_1,\mathcal{L}_2,\overline{\mathcal{L}}_5}$ and $V_{\mathcal{L}_3,\mathcal{L}_4,\mathcal{L}_5}$ and outputs a state in the Hilbert space associated with the boundary gray circle. The output state should have conformal weights $(h, \tilde{h}) = (0,0)$ and hence live in the junction vector space $V_{\mathcal{L}_1,\mathcal{L}_2,\mathcal{L}_3,\mathcal{L}_4}$.

- **Modular covariance:** The modular covariance of the correlation functions of local operators on Riemann surfaces carries over to those with TDLs and defect operators.

  Let us focus on the partition function on a torus $T^2$. When a TDL $\mathcal{L}$ is inserted along the time direction of $T^2$, it modifies the quantization by a twisted boundary condition on the space circle, which defines the defect Hilbert space $\mathcal{H}_{\mathcal{L}}$. The torus partition function is then evaluated by a trace over $\mathcal{H}_{\mathcal{L}}$,

$$Z_{\mathcal{L}}(\tau, \bar{\tau}) = \operatorname{Tr}_{\mathcal{H}_{\mathcal{L}}} \left( q^{L_0 - \frac{c}{24}} \bar{q}^{\tilde{L}_0 - \frac{c}{24}} \right). \quad\quad (1.13)$$

  After a modular S-transformation, the TDL $\mathcal{L}$ becomes inserting along the space direction and implements the $\widehat{\mathcal{L}}$ action on the Hilbert space $\mathcal{H}$ of local operators. This gives the twisted partition function $Z^{\mathcal{L}}(\tau, \bar{\tau}) = \operatorname{Tr}_{\mathcal{H}}(\widehat{\mathcal{L}} q^{L_0 - \frac{c}{24}} \bar{q}^{\tilde{L}_0 - \frac{c}{24}})$. In summary, the modular invariance requires that

$$\operatorname{Tr}_{\mathcal{H}_{\mathcal{L}}} \left( q^{L_0 - \frac{c}{24}} \bar{q}^{\tilde{L}_0 - \frac{c}{24}} \right) = Z_{\mathcal{L}}(\tau, \bar{\tau}) = Z^{\mathcal{L}}\left( -\frac{1}{\tau}, -\frac{1}{\bar{\tau}} \right) = \operatorname{Tr}_{\mathcal{H}} \left( \widehat{\mathcal{L}} \tilde{q}^{L_0 - \frac{c}{24}} \bar{\tilde{q}}^{\tilde{L}_0 - \frac{c}{24}} \right), \quad (1.14)$$

where $\tilde{q} = e^{-\frac{2\pi i}{\tau}}$.

# 2 TDLs in fermionic CFTs

## 2.1 Defining properties

The TDLs in fermionic CFTs obey all the defining properties stated in Section 2 of [8] and reviewed in Section 1.1, except the following modifications and additions.

**1. (Fermionic defect operator)** The defect Hilbert space at the junctions of TDLs admits an $\mathbb{Z}_2$ grading $\sigma$, which assigns 0 to bosonic and 1 to fermionic states or defect operators. The fermionic defect operators are Grassmann-valued, i.e. exchanging the order of two fermionic defect operators inside a correlation function acquires a minus sign. Hence, we add an extra integer label (in red color) to the junctions to specify their ordering inside correlation functions. The junction vector space $V_{\mathcal{L}_1,\cdots,\mathcal{L}_n}$ inherits the $\mathbb{Z}_2$ grading of the defect Hilbert space $\mathcal{H}_{\mathcal{L}_1,\cdots,\mathcal{L}_n}$. We denote the bosonic subspace by $V^{\mathrm{b}}_{\mathcal{L}_1,\cdots,\mathcal{L}_n}$ and the fermionic subspace by $V^{\mathrm{f}}_{\mathcal{L}_1,\cdots,\mathcal{L}_n}$. For example, the correlation function involving the fermionic junction vectors $\psi \in V^{\mathrm{f}}_{\mathcal{L}_1,\mathcal{L}_2,\overline{\mathcal{L}}_5}$ and $\psi' \in V^{\mathrm{f}}_{\mathcal{L}_5,\mathcal{L}_3,\mathcal{L}_4}$ obeys[7]

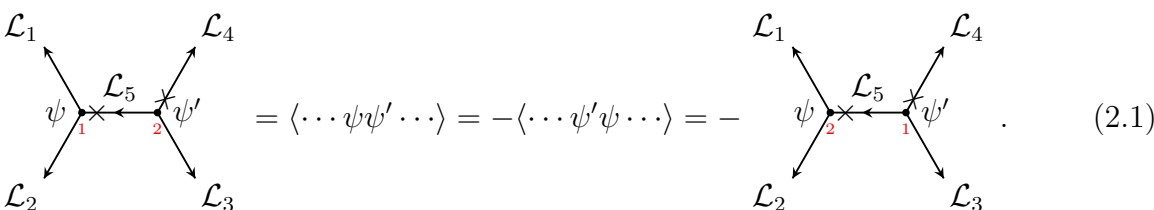

$$= \langle \cdots \psi\psi' \cdots \rangle = -\langle \cdots \psi'\psi \cdots \rangle = - \qquad . \qquad (2.1)$$

On Riemann surfaces, defect (or bulk local) operators satisfy the Neveu-Schwarz (NS) or Ramond (R) boundary condition. More precisely, an operator is identified with $(-1)^{\sigma\nu}$ times its image under a $2\pi$ shift along a non-contractible cycle on a Riemann surface, where $\sigma$ is the $\mathbb{Z}_2$ grading of the operator and $\nu = 1$ ($\nu = 0$) for the NS (R) boundary condition. Let us focus on the NS boundary condition. For example, consider on a cylinder a pair of defect operators $\mathcal{O}$'s connected by a TDL $\mathcal{L}$ that wraps around the cylinder once. On the

---

[7]Throughout this paper, all the correlation functions are in the Euclidean path integral formalism.

double cover of the cylinder, we have

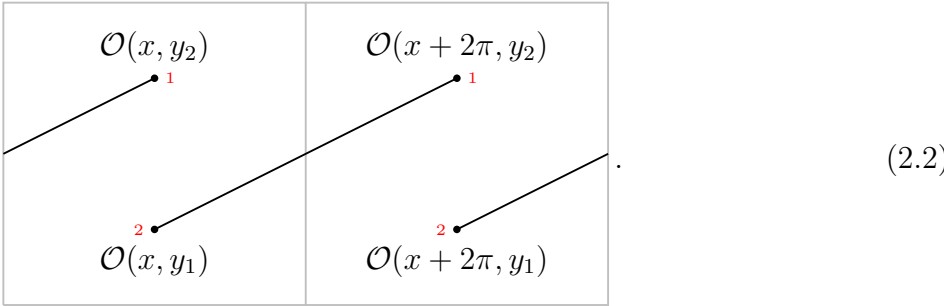

$$(2.2)$$

The operators on different sheets are identified as

$$\mathcal{O}(x + 2\pi, y_i) = (-1)^\sigma \mathcal{O}(x, y_i). \tag{2.3}$$

By the state/operator correspondence, this rule implies that a $2\pi$ rotation of a defect operator produces a phase $e^{2\pi i(s+\frac{1}{2}\sigma)}$ that depends on both the spin $s$ and the $\mathbb{Z}_2$ grading $\sigma$ of the defect operator.

We will assume the NS boundary condition for all the non-contractible cycles throughout this paper. The correlation functions with the R boundary condition for some or all non-contractible cycles can be obtained by inserting the fermion parity TDL $(-1)^F$ wrapping the dual cycles.

**2. (Simple TDL)** A TDL $\mathcal{L}$ is simple if the junction vector space $V_{\mathcal{L},\overline{\mathcal{L}}}$ is isomorphic to either $\mathbb{C}^{1|0}$ or $\mathbb{C}^{1|1}$. The former TDL is called *m-type*, and the later is called *q-type*. For a q-type TDL $\mathcal{L}_q$, the fermionic state in the junction vector space $V_{\mathcal{L}_q,\overline{\mathcal{L}}_q} \cong \mathbb{C}^{1|1}$ corresponds to a topological fermionic defect operator $\psi$ (with conformal weight $(h, \tilde{h}) = (0, 0)$) living on the worldline of $\mathcal{L}_q$. We will refer to $\psi$ as the 1d Majorana fermion.

On an oriented q-type TDL $\mathcal{L}$, the 1d Majorana fermion $\psi$ can be pair-created, and by a suitable choice of the normalization of $\psi$ we have

$$(2.4)$$

On an unoriented q-type TDL $\mathcal{L}$, there is no canonical way to completely fix the pair-creation

of $\psi$, and we have[8],[9]

$$
\left| \ \right| = \begin{array}{c} {\scriptstyle 1} \bullet \ \psi \\ {\scriptstyle 2} \bullet \ \psi \end{array} \qquad \text{or} \qquad \left| \ \right| = \begin{array}{c} {\scriptstyle 2} \bullet \ \psi \\ {\scriptstyle 1} \bullet \ \psi \end{array} = - \begin{array}{c} {\scriptstyle 1} \bullet \ \psi \\ {\scriptstyle 2} \bullet \ \psi \end{array} .
\tag{2.5}
$$

For a junction involving a q-type TDL $\mathcal{L}$, the 1d Majorana fermion on $\mathcal{L}$ induces an $\mathbb{Z}_2$-odd map $\Psi$ on the defect Hilbert space associated with the junction, which preserves the Virasoro algebra and squares to the identity map due to (2.4) or (2.5). It follows that the spectrum of the defect Hilbert space is doubly degenerate. For example, consider a trivalent junction of a q-type TDL $\mathcal{L}_3$ and two other TDLs $\mathcal{L}_1$ and $\mathcal{L}_2$, the 1d Majorana fermion $\psi$ on $\mathcal{L}_3$ can act on the junction vector space $V_{\mathcal{L}_1, \mathcal{L}_2, \mathcal{L}_3}$ as

$$
\tag{2.6}
$$

where $\Psi_{\mathcal{L}_3} : V_{\mathcal{L}_1, \mathcal{L}_2, \mathcal{L}_3} \to V_{\mathcal{L}_1, \mathcal{L}_2, \mathcal{L}_3}$ is a $\mathbb{Z}_2$-odd linear map, that squares to the identity map, i.e. $\Psi_{\mathcal{L}_3}^2 = 1$.[10]

**3. (H-junction and partial fusion)** By the locality property, an H-junction involving four external simple TDLs $\mathcal{L}_1, \cdots, \mathcal{L}_4$ and an internal simple TDL $\mathcal{L}_5$ is a bilinear map

$$
H^{\mathcal{L}_1, \mathcal{L}_4}_{\mathcal{L}_2, \mathcal{L}_3}(\mathcal{L}_5) \equiv \qquad : \qquad V_{\mathcal{L}_1, \mathcal{L}_2, \overline{\mathcal{L}}_5} \otimes V_{\mathcal{L}_3, \mathcal{L}_4, \mathcal{L}_5} \to V_{\mathcal{L}_1, \mathcal{L}_2, \mathcal{L}_3, \mathcal{L}_4} ,
\tag{2.7}
$$

which takes two junction vectors in the junction vector spaces $V_{\mathcal{L}_1, \mathcal{L}_2, \overline{\mathcal{L}}_5}$ and $V_{\mathcal{L}_3, \mathcal{L}_4, \mathcal{L}_5}$, and produces a state on the gray circle in the junction vector space $V_{\mathcal{L}_1, \mathcal{L}_2, \mathcal{L}_3, \mathcal{L}_4}$. The ordering of the junction vector spaces in the tensor product corresponds to the ordering of the junction

---

[8]Throughout this paper, our discussions on the unoriented TDLs are always on flat manifolds. Hence, we ignore any potential orientation reversal anomaly.

[9]Our convention of the pair-creations (2.4) and (2.5) is related to the one in [72] by a field redefinition $\psi \to e^{\pm \frac{\pi i}{4}} \psi$.

[10]For related discussions, see [72, 45, 82].

vectors in correlation functions, and is determined by the red integer labels in the graph (2.7).

If the internal TDL $\mathcal{L}_5$ is a q-type TDL, a pair of 1d Majorana fermions can be pair-created and act on the junction vector spaces $V_{\mathcal{L}_1,\mathcal{L}_2,\overline{\mathcal{L}}_5}$ and $V_{\mathcal{L}_3,\mathcal{L}_4,\mathcal{L}_5}$, explicitly as

which can be equivalently written as

$$H^{\mathcal{L}_1,\mathcal{L}_4}_{\mathcal{L}_2,\mathcal{L}_3}(\mathcal{L}_5) = H^{\mathcal{L}_1,\mathcal{L}_4}_{\mathcal{L}_2,\mathcal{L}_3}(\mathcal{L}_5) \circ P_{\mathcal{L}_5} , \tag{2.9}$$

where $P_{\mathcal{L}}$ is a projection map

$$P_{\mathcal{L}} \equiv \begin{cases} 1 & \text{for m-type } \mathcal{L}, \\ \frac{1}{2}(1 + \Psi_{\overline{\mathcal{L}}} \otimes \Psi_{\mathcal{L}}) & \text{for q-type } \mathcal{L}, \end{cases} \tag{2.10}$$

and the "1" stands for the identity map.[11]

On a local patch, a pair of TDLs can be partially fused to a TDL $\mathcal{L}_1\mathcal{L}_2$, with a set of junction vectors $v_i \in V_{\mathcal{L}_1,\mathcal{L}_2,\overline{\mathcal{L}_1\mathcal{L}_2}}$ and $\tilde{v}_i \in V_{\overline{\mathcal{L}}_2,\overline{\mathcal{L}}_1,\mathcal{L}_1\mathcal{L}_2}$ inserted at the trivalent junctions,

The combination $\sum_i v_i \otimes \tilde{v}_i$ is unique in the projection

$$\bigoplus_{\mathcal{L}_i \subset \mathcal{L}_1\mathcal{L}_2} P_{\mathcal{L}_i}\left(V_{\mathcal{L}_1,\mathcal{L}_2,\overline{\mathcal{L}}_i} \otimes V_{\overline{\mathcal{L}}_2,\overline{\mathcal{L}}_1,\mathcal{L}_i}\right) . \tag{2.12}$$

## 2.2   Corollaries

Most of the corollaries in Section 2 of [8] hold for TDLs in fermionic CFTs. We discuss some modifications and additional corollaries.

---

[11]When $\mathcal{L}_5$ is unoriented, the two different choices of the pair-creation in (2.5) give projection operators onto different linearly independent subspaces.

### 2.2.1 F-move

The direct sum of all possible H-junctions with external simple TDLs $\mathcal{L}_1, \cdots, \mathcal{L}_4$ gives a map $H_{\mathcal{L}_2,\mathcal{L}_3}^{\mathcal{L}_1,\mathcal{L}_4} \equiv \bigoplus_{\text{simple } \mathcal{L}_5} H_{\mathcal{L}_2,\mathcal{L}_3}^{\mathcal{L}_1,\mathcal{L}_4}(\mathcal{L}_5)$. By (2.9), $H_{\mathcal{L}_2,\mathcal{L}_3}^{\mathcal{L}_1,\mathcal{L}_4}$ satisfies

$$H_{\mathcal{L}_2,\mathcal{L}_3}^{\mathcal{L}_1,\mathcal{L}_4} = H_{\mathcal{L}_2,\mathcal{L}_3}^{\mathcal{L}_1,\mathcal{L}_4} \circ \prod_{\text{simple } \mathcal{L}_5} P_{\mathcal{L}_5}. \tag{2.13}$$

After restricting the domain of $H_{\mathcal{L}_2,\mathcal{L}_3}^{\mathcal{L}_1,\mathcal{L}_4}$ to the projection

$$\bigoplus_{\text{simple } \mathcal{L}_5} P_{\mathcal{L}_5}\left(V_{\mathcal{L}_1,\mathcal{L}_2,\overline{\mathcal{L}}_5} \otimes V_{\mathcal{L}_3,\mathcal{L}_4,\mathcal{L}_5}\right), \tag{2.14}$$

one can find the inverse map $\overline{H}_{\mathcal{L}_2,\mathcal{L}_3}^{\mathcal{L}_1,\mathcal{L}_4}$ following the same manipulation as in [8] using the partial fusion (2.11), i.e. the composition $\overline{H}_{\mathcal{L}_2,\mathcal{L}_3}^{\mathcal{L}_1,\mathcal{L}_4} \circ H_{\mathcal{L}_2,\mathcal{L}_3}^{\mathcal{L}_1,\mathcal{L}_4}$ equals to the identity map on (2.14). One also define $\overline{H}_{\mathcal{L}_2,\mathcal{L}_3}^{\mathcal{L}_1,\mathcal{L}_4}(\mathcal{L}_6)$ as the projection of $\overline{H}_{\mathcal{L}_2,\mathcal{L}_3}^{\mathcal{L}_1,\mathcal{L}_4}$ onto the subspace $P_{\mathcal{L}_6}(V_{\mathcal{L}_1,\mathcal{L}_2,\overline{\mathcal{L}}_6} \otimes V_{\mathcal{L}_3,\mathcal{L}_4,\mathcal{L}_6})$.

The H-junction crossing kernel $\widetilde{K}$ is define as the composition[12,13]

$$\widetilde{K}_{\mathcal{L}_2,\mathcal{L}_3}^{\mathcal{L}_1,\mathcal{L}_4}(\mathcal{L}_5,\mathcal{L}_6) \equiv C_{\mathcal{L}_4,\mathcal{L}_1,\mathcal{L}_6} \circ \overline{H}_{\mathcal{L}_3,\mathcal{L}_4}^{\mathcal{L}_2,\mathcal{L}_1}(\mathcal{L}_6) \circ C_{\mathcal{L}_1,\mathcal{L}_2,\mathcal{L}_3,\mathcal{L}_4} \circ H_{\mathcal{L}_2,\mathcal{L}_3}^{\mathcal{L}_1,\mathcal{L}_4}(\mathcal{L}_5) \circ C_{\mathcal{L}_5,\mathcal{L}_3,\mathcal{L}_4}$$
$$: P_{\mathcal{L}_5}(V_{\mathcal{L}_1,\mathcal{L}_2,\overline{\mathcal{L}}_5} \otimes V_{\mathcal{L}_5,\mathcal{L}_3,\mathcal{L}_4}) \to P_{\mathcal{L}_6}(V_{\mathcal{L}_2,\mathcal{L}_3,\overline{\mathcal{L}}_6} \otimes V_{\mathcal{L}_1,\mathcal{L}_6,\mathcal{L}_4}). \tag{2.15}$$

The F-symbol $F_{\mathcal{L}_4}^{\mathcal{L}_1,\mathcal{L}_2,\mathcal{L}_3}$ is defined as the inverse of the direct sum of the crossing kernel as

$$F_{\mathcal{L}_4}^{\mathcal{L}_1,\mathcal{L}_2,\mathcal{L}_3} \equiv \left(\bigoplus_{\text{simple } \mathcal{L}_5, \mathcal{L}_6} \widetilde{K}_{\mathcal{L}_2,\mathcal{L}_1}^{\mathcal{L}_3,\overline{\mathcal{L}}_4}(\mathcal{L}_6,\mathcal{L}_5)\right)^{-1}. \tag{2.16}$$

Further projecting $F_{\mathcal{L}_4}^{\mathcal{L}_1,\mathcal{L}_2,\mathcal{L}_3}$ to the subspace $\text{Hom}(P_{\mathcal{L}_5}(V_{\mathcal{L}_2,\mathcal{L}_1,\overline{\mathcal{L}}_5} \otimes V_{\mathcal{L}_3,\mathcal{L}_5,\overline{\mathcal{L}}_4}), P_{\mathcal{L}_6}(V_{\mathcal{L}_3,\mathcal{L}_2,\overline{\mathcal{L}}_6} \otimes V_{\mathcal{L}_6,\mathcal{L}_1,\overline{\mathcal{L}}_4}))$ gives $F_{\mathcal{L}_4}^{\mathcal{L}_1,\mathcal{L}_2,\mathcal{L}_3}(\mathcal{L}_5,\mathcal{L}_6)$. Graphically, the F-symbol $F_{\mathcal{L}_4}^{\mathcal{L}_1,\mathcal{L}_2,\mathcal{L}_3}(\mathcal{L}_5,\mathcal{L}_6)$ gives the F-move

$$\tag{2.17}$$

---

[12]For notation simplicity, here we abuse the notation a bit by denoting the conjugation of the projection map $P_{\mathcal{L}}$ (by cyclic permutation maps) also as $P_{\mathcal{L}}$.

[13]We restrict the domain of the crossing kernel $\widetilde{K}_{\mathcal{L}_2,\mathcal{L}_3}^{\mathcal{L}_1,\mathcal{L}_4}(\mathcal{L}_5,\mathcal{L}_6)$ to be the projection $P_{\mathcal{L}_5}(V_{\mathcal{L}_1,\mathcal{L}_2,\overline{\mathcal{L}}_5} \otimes V_{\mathcal{L}_5,\mathcal{L}_3,\mathcal{L}_4})$, because in the following we would like to consider the inverse of the crossing kernel $\widetilde{K}_{\mathcal{L}_2,\mathcal{L}_3}^{\mathcal{L}_1,\mathcal{L}_4}$.

Here, in the remaining of this section, and in Section 3, we adopt the no ×-mark convention such that the lines pointing down (up) from a junction correspond to the lines with (without) the ×-marks.

It would be useful to regard the F-symbol $F_{\mathcal{L}_4}^{\mathcal{L}_1,\mathcal{L}_2,\mathcal{L}_3}(\mathcal{L}_5,\mathcal{L}_6)$ as a map between $V_{\mathcal{L}_2,\mathcal{L}_1,\overline{\mathcal{L}}_5} \otimes V_{\mathcal{L}_3,\mathcal{L}_5,\overline{\mathcal{L}}_4}$ and $V_{\mathcal{L}_3,\mathcal{L}_2,\overline{\mathcal{L}}_6} \otimes V_{\mathcal{L}_6,\mathcal{L}_1,\overline{\mathcal{L}}_4}$ that satisfies the projection condition[14]

$$F_{\mathcal{L}_4}^{\mathcal{L}_1,\mathcal{L}_2,\mathcal{L}_3}(\mathcal{L}_5,\mathcal{L}_6) = P_{\mathcal{L}_6} \circ F_{\mathcal{L}_4}^{\mathcal{L}_1,\mathcal{L}_2,\mathcal{L}_3}(\mathcal{L}_5,\mathcal{L}_6) \circ P_{\mathcal{L}_5}\,. \tag{2.18}$$

We give the solution to the projection condition in the matrix representation in Appendix A.

Finally, we note that the F-move is a local manipulation of TDLs, i.e. it is performed in a local patch; hence should be independent of the boundary conditions of the fermionic operators.

### 2.2.2 Super pentagon identity

As illustrated in the following commutative diagram

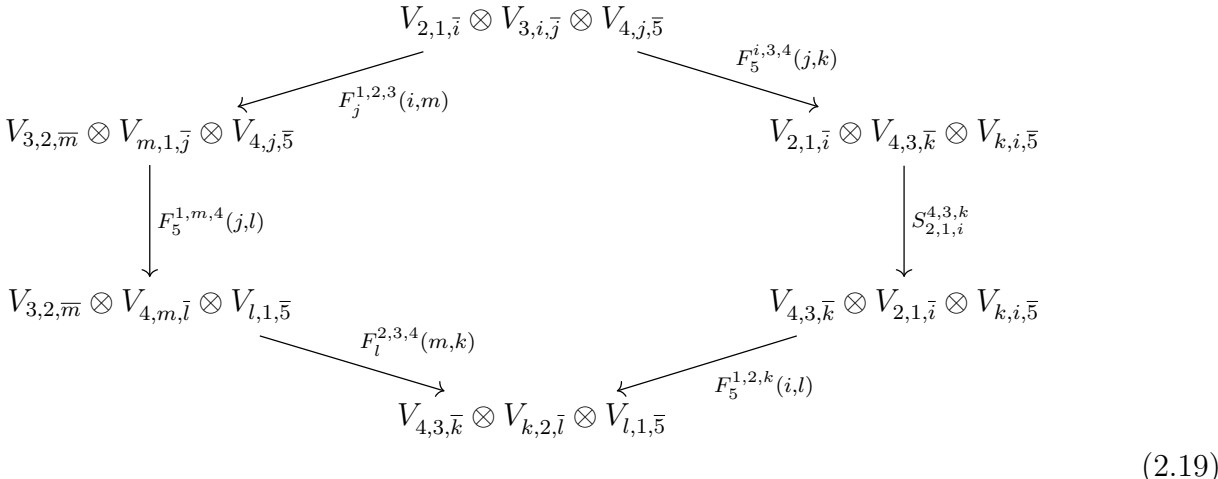

$$\tag{2.19}$$

the F-move satisfies the super pentagon identity

$$F_5^{1,2,k}(i,l) \circ S_{2,1,i}^{4,3,k} \circ F_5^{i,3,4}(j,k) = \sum_m F_l^{2,3,4}(m,k) \circ F_5^{1,m,4}(j,l) \circ F_j^{1,2,3}(i,m)\,, \tag{2.20}$$

where we abbreviated $F_{\mathcal{L}_4}^{\mathcal{L}_1,\mathcal{L}_2,\mathcal{L}_3}(\mathcal{L}_5,\mathcal{L}_6)$ by $F_4^{1,2,3}(5,6)$. The map $S_{2,1,i}^{4,3,k}$ exchanges the ordering of a pair of junction vectors. More precisely, we have $S_{1,2,3}^{4,5,6} : V_{\mathcal{L}_1,\mathcal{L}_2,\overline{\mathcal{L}}_3} \otimes V_{\mathcal{L}_4,\mathcal{L}_5,\overline{\mathcal{L}}_6} \to V_{\mathcal{L}_4,\mathcal{L}_5,\overline{\mathcal{L}}_6} \otimes V_{\mathcal{L}_1,\mathcal{L}_2,\overline{\mathcal{L}}_3}$ defined by

$$S_{1,2,3}^{4,5,6}(v \otimes v') = (-1)^{\sigma(v)\sigma(v')} v' \otimes v\,, \tag{2.21}$$

---

[14]Our treatment of the F-moves differs from [78] that we impose the projection condition (2.18) instead of the projective unitary condition (26) in [78].

where $\sigma$ is the $\mathbb{Z}_2$-grading of the junction vector spaces.

We emphasize that when any of the internal simple TDLs is q-type, both hand sides of the super pentagon identity are maps between the projections of the tensor products of the junction vector spaces, $P_{\mathcal{L}_i} P_{\mathcal{L}_j}(V_{\mathcal{L}_2, \mathcal{L}_1, \mathcal{L}_i} \otimes V_{\mathcal{L}_3, \mathcal{L}_i, \mathcal{L}_j} \otimes V_{\mathcal{L}_4, \mathcal{L}_j, \mathcal{L}_5})$ and $P_{\mathcal{L}_k} P_{\mathcal{L}_l}(V_{\mathcal{L}_3, \mathcal{L}_4, \mathcal{L}_k} \otimes V_{\mathcal{L}_k, \mathcal{L}_2, \mathcal{L}_l} \otimes V_{\mathcal{L}_l, \mathcal{L}_1, \mathcal{L}_5})$.

Similar to the bosonic case where the anomalies for global symmetries $G$/invertible TDLs can be classified by the H-junction crossing relations that solve the pentagon identity, t 'Hooft anomalies in a 2d fermionic CFT can also be classified in the supe pentagon identity. To see that, recall that it was known [83, 74] that this type of anomalies in a 2d fermionic system can be captured by the group $\mathrm{Hom}(\Omega_3^{\mathrm{spin}}(BG), U(1))$. This group basically can be described by a triples of chain $(w, p, a)$, with $w \in C^3(BG, U(1))$, $p$ a cocycle in $Z^2(BG, \mathbb{Z}_2)$ and $a$ a cocycle in $Z^1(BG, \mathbb{Z}_2)$ [76]. In terms of the data of the super fusion category, $a$ encodes the information whether a TDL has a 1d Majorana fermion, $p$ specifies the $\mathbb{Z}_2$ grading $\sigma$ on a trivalent junction, and $w$ encodes the H-junction crossing relation [84].

### 2.2.3 Universal sector

The F-moves of the H-junctions with no more than two non-trivial external TDLs form a simple set of data that is universal for all the TDLs, and can be solved by the super pentagon equations with also no more than two non-trivial external TDLs. We will call it the universal sector.

For the m-type TDLs, the analysis of the universal sector is identical to the TDLs in bosonic CFTs [8], and we review it here. For simplicity, we consider unoriented m-type TDLs. First, let us choose some convenient gauge to fix some F-moves. Note that for the following trivalent junctions, we can perform the following "gauge transformations",

$$\includegraphics{eq2.22} \qquad \rightarrow \quad x \quad , \qquad \rightarrow \quad y \quad , \qquad (2.22)$$

where the dotted lines denote the trivial TDL $I$. By using these, we can always normalize the F-moves of the following graphs,

$$= \quad , \qquad = \quad . \qquad (2.23)$$

With the above normalization, one can further focus on all super pentagon equations with the F-moves of H-junctions containing only two non-trivial external m-type TDLs. The solution is unique, and all these F-symbols are represented by $1 \times 1$ identity matrices.

Next, we focus on the universal sector of an unoriented q-type TDL $\mathcal{L}$. The result for the universal sector of oriented q-type TDLs will be summarized in Appendix B. The junction vector space of a trivalent junction involving two $\mathcal{L}$'s and a trivial TDL $I$ is isomorphic to $\mathbb{C}^{1|1}$. More precisely, we have three isomorphic junction vector spaces

$$V_{I,\mathcal{L},\mathcal{L}} \cong V_{\mathcal{L},I,\mathcal{L}} \cong V_{\mathcal{L},\mathcal{L},I} \cong \mathbb{C}^{1|1} \,. \tag{2.24}$$

The 1d Majorana fermion on $\mathcal{L}$ acts on these junction vector spaces as

where the dotted lines denote the trivial TDL $I$, and the junctions with (or without) a black dot are associated with the fermionic (or bosonic) junction vector subspaces.

Changing the basis of the junction vector spaces gives the "gauge transformations"

The gauge symmetry can be partially fixed by the gauge conditions

$$\alpha_1 = \frac{1}{\alpha_2} \equiv \alpha \,, \quad \beta_1 = \frac{1}{\beta_2} \equiv \beta \,, \quad \gamma_1 = \frac{1}{\gamma_2} \equiv \gamma \,. \tag{2.27}$$

The above gauge conditions are invariant under the residual gauge transformations given by (2.26) with the constraints

$$x_1 = \pm x_2 \,, \quad y_1 = \pm y_2 \,, \quad z_1 = \pm z_2 \,. \tag{2.28}$$

Let us focus on the F-symbols with only two non-trivial external TDLs

$$F_I^{\mathcal{L},\mathcal{L},I}(I,\mathcal{L}) \,, \; F_I^{\mathcal{L},I,\mathcal{L}}(\mathcal{L},\mathcal{L}) \,, \; F_\mathcal{L}^{\mathcal{L},I,I}(\mathcal{L},I) \,, \; F_I^{I,\mathcal{L},\mathcal{L}}(\mathcal{L},I) \,, \; F_\mathcal{L}^{I,\mathcal{L},I}(\mathcal{L},\mathcal{L}) \,, \; F_\mathcal{L}^{I,I,\mathcal{L}}(I,\mathcal{L}) \,. \tag{2.29}$$

The F-symbols will be regarded as maps between the unprojected junction vector spaces that satisfy the projection condition (2.18), and represented by $4 \times 4$ matrices $\mathcal{F}$'s. The projection matrices can be computed using the 1d Majorana fermion actions (2.25). For example, the projection matrix involved in the projection condition of the F-matrix $\mathcal{F}_I^{\mathcal{L},\mathcal{L},I}(I,\mathcal{L})$ are given by the relations

$$(2.30)$$

where $\epsilon_1 = \pm 1$ is the sign ambiguity of the pair-creation on unoriented q-type TDLs (2.5). Similarly, we could compute the projection matrices for the other F-matrices. There is a sign ambiguity for each pair-creation, and we need to introduce seven other more undetermined signs $\epsilon_2$, $\epsilon_2'$, $\epsilon_3$, $\epsilon_4$, $\epsilon_5$, $\epsilon_5'$, $\epsilon_6 = \pm 1$.

Solving the projection conditions as in Appendix A, we find the following form of the F-matrices

$$
\mathcal{F}_I^{\mathcal{L},\mathcal{L},I}(I,\mathcal{L}) = \begin{pmatrix} f_{1b} & 0 \\ 0 & f_{1f} \end{pmatrix} \begin{pmatrix} 1 & \frac{\epsilon_1 \gamma}{\alpha} & 0 & 0 \\ 0 & 0 & 1 & \alpha\gamma \end{pmatrix}, \quad
\mathcal{F}_I^{\mathcal{L},I,\mathcal{L}}(\mathcal{L},\mathcal{L}) = \begin{pmatrix} 1 & 0 \\ \epsilon_2\alpha\gamma & 0 \\ 0 & 1 \\ 0 & \frac{\gamma}{\alpha} \end{pmatrix} \begin{pmatrix} f_{2b} & 0 \\ 0 & f_{2f} \end{pmatrix} \begin{pmatrix} 1 & \frac{\epsilon_2' \gamma}{\beta} & 0 & 0 \\ 0 & 0 & 1 & \beta\gamma \end{pmatrix},
$$

$$
\mathcal{F}_{\mathcal{L}}^{\mathcal{L},I,I}(\mathcal{L},I) = \begin{pmatrix} 1 & 0 \\ \epsilon_3 & 0 \\ 0 & 1 \\ 0 & \frac{1}{\alpha^2} \end{pmatrix} \begin{pmatrix} f_{3b} & 0 \\ 0 & f_{3f} \end{pmatrix}, \quad
\mathcal{F}_I^{I,\mathcal{L},\mathcal{L}}(\mathcal{L},I) = \begin{pmatrix} 1 & 0 \\ \epsilon_4\beta\gamma & 0 \\ 0 & 1 \\ 0 & \frac{\gamma}{\beta} \end{pmatrix} \begin{pmatrix} f_{4b} & 0 \\ 0 & f_{4f} \end{pmatrix},
$$

$$
\mathcal{F}_{\mathcal{L}}^{I,\mathcal{L},I}(\mathcal{L},\mathcal{L}) = \begin{pmatrix} 1 & 0 \\ \frac{\epsilon_5\beta}{\alpha} & 0 \\ 0 & 1 \\ 0 & \frac{1}{\beta\alpha} \end{pmatrix} \begin{pmatrix} f_{5b} & 0 \\ 0 & f_{5f} \end{pmatrix} \begin{pmatrix} 1 & \frac{\epsilon_5'\beta}{\alpha} & 0 & 0 \\ 0 & 0 & 1 & \alpha\beta \end{pmatrix}, \quad
\mathcal{F}_{\mathcal{L}}^{I,I,\mathcal{L}}(I,\mathcal{L}) = \begin{pmatrix} f_{6b} & 0 \\ 0 & f_{6f} \end{pmatrix} \begin{pmatrix} 1 & \epsilon_6 & 0 & 0 \\ 0 & 0 & 1 & \beta^2 \end{pmatrix}.
$$

$$(2.31)$$

The residual gauge symmetry ((2.26) with (2.28)) can be used to fix

$$
f_{1b} = f_{2b} = \frac{1}{2}. \tag{2.32}
$$

There are ten super pentagon equations with only two non-trivial external TDLs. Solving

them, we find two gauge inequivalent solutions of the F-moves,

$$\alpha = \beta = 1 \,, \quad f_{1f} = \frac{1}{2}\gamma^{-1} \,, \quad f_{2f} = -\frac{\gamma^2}{2} \,,$$

$$f_{3b} = f_{3f} = f_{4b} = 1 \,, \quad f_{4f} = \gamma^{-1} \,, \quad f_{5b} = f_{5f} = f_{6b} = f_{6f} = \frac{1}{2} \,, \tag{2.33}$$

$$\epsilon_1 = \epsilon_2 = \epsilon_2' = \epsilon_3 = \epsilon_4 = \epsilon_5 = \epsilon_5' = \epsilon_6 = 1 \,,$$

for $\gamma = e^{\frac{i\pi}{4}}, e^{\frac{3i\pi}{4}}$. Note that the ambiguities $\epsilon_1, \cdots, \epsilon_6$ are completely fixed. This effectively gives the unoriented TDL $\mathcal{L}$ an orientation such that all the lines in (2.25) (and also (2.30)) have an upward pointing orientation.

**Constraints on the projection matrices**  Since the 1d Majorana fermions can freely move along the q-type TDLs, there are additional constraints on the actions (2.25), as shown in the following graphs,

$$\tag{2.34}$$

where we have used eq. (2.5) and (2.25) for the unoriented q-type line. Therefore we have the constraints on the $\alpha$, $\beta$ and $\gamma$ as

$$\alpha^4 = 1 \,, \quad \beta^4 = 1 \,, \quad \text{and} \quad \gamma^4 = -1 \,. \tag{2.35}$$

One can check that our universal sector solutions (2.33) indeed satisfy the above constraints.[15]

### 2.2.4    Rotation on defect operators

As discussed in the previous section, there are phases and signs producing from rotating the defect operators, switching the order of the defect operators in correlation functions, and moving the defect operators around non-contractible cycles. They are consistent with each other as we now see. Consider a defect operator $\mathcal{O}$ in the defect Hilbert space $\mathcal{H}_{\mathcal{L}}$ of an

---

[15]There are actually sign ambiguities in the relations (2.34), but the universal sector solutions (2.33) fix those sign ambiguities.

unoriented TDL $\mathcal{L}$, which can be normalized such that the two-point function of a pair of $\mathcal{O}$'s connected by $\mathcal{L}$ is

$$\overset{\underset{1}{\bullet} \qquad \underset{2}{\bullet}}{\mathcal{O}(z_1, \bar{z}_1) \qquad \mathcal{O}(z_2, \bar{z}_2)} = z_{12}^{-2h} \bar{z}_{12}^{-2\tilde{h}} . \tag{2.36}$$

Bringing $z_2$ to $z_1$ and $z_1$ to $z_2$ counterclockwisly gives

$$\overset{\mathcal{O}(z_1, \bar{z}_1) \qquad \mathcal{O}(z_2, \bar{z}_2)}{\underset{2}{\bullet} \qquad \underset{1}{\bullet}} = e^{-2\pi i s} z_{12}^{-2h} \bar{z}_{12}^{-2\tilde{h}} , \tag{2.37}$$

where the phase on the RHS is given by taking $z_{12} \to e^{i\pi} z_{12}$. Rotating the left operator by degree $2\pi$ gives a phase $e^{2\pi i(s+\frac{\sigma}{2})}$, and we have

$$\overset{\underset{2}{\bullet} \qquad \underset{1}{\bullet}}{\mathcal{O}(z_1, \bar{z}_1) \qquad \mathcal{O}(z_2, \bar{z}_2)} = e^{\pi i \sigma} z_{12}^{-2h} \bar{z}_{12}^{-2\tilde{h}} . \tag{2.38}$$

We see that the resulting phase $e^{\pi i \sigma}$ is consistent with the sign given by exchanging the order of the two defect operators in the two-point function.

To see the consistency between the rotation phase and the boundary condition of the defect operators, let us consider a pair of 1d Majorana fermions $\psi$ on antipodal points of a q-type TDL $\mathcal{L}$ inserted on a unit circle centered at the origin of a complex plane with complex coordinate $z$. Under the exponential map $z = e^{iw}$, the complex plane is mapped to an infinite cylinder with the NS boundary condition for the non-contractible cycle,

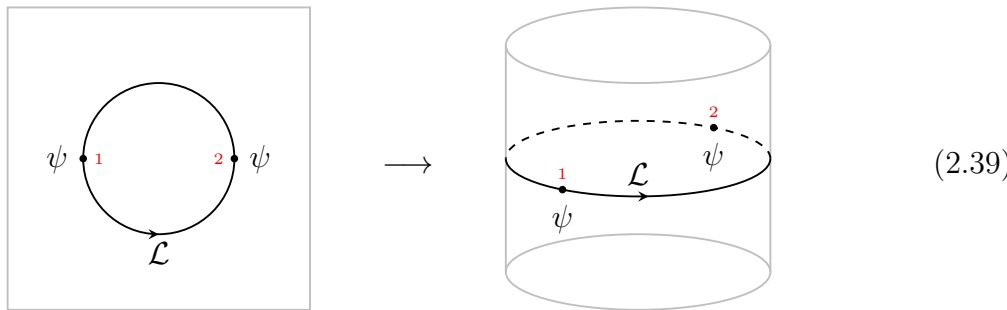

$$\tag{2.39}$$

Now, moving both of the two $\psi$'s on the cylinder counterclockwisly by half of the circle gives a minus sign due to the NS-boundary condition. On the plane, it corresponds to first moving

the $\psi$'s to the middle configuration in (2.40) then rotating the two $\psi$'s by degree $\pi$.

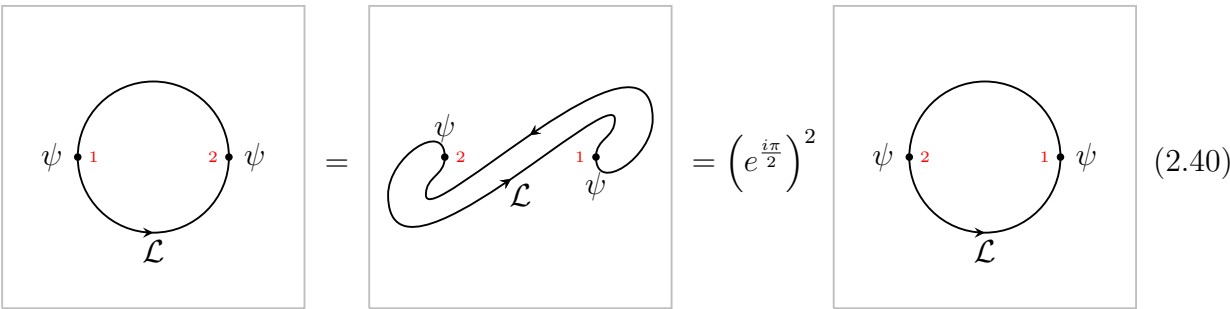

$$\psi \overset{1}{\bullet} \quad \overset{2}{\bullet} \psi \quad = \quad \psi \quad \mathcal{L} \quad = \left( e^{\frac{i\pi}{2}} \right)^2 \quad \psi \overset{2}{\bullet} \quad \overset{1}{\bullet} \psi \tag{2.40}$$

The rotation produces the phase $(e^{\frac{i\pi}{2}})^2$ which matches with the previous minus sign in the cylinder case. By a similar manipulation, we also find

$$\psi \overset{}{\bullet} \quad = 0 \, , \tag{2.41}$$

which was discussed in [72].[16]

### 2.2.5 Fusion coefficients

The fusion coefficients are determined by the dimensions of the junction vector spaces as

$$\mathcal{L}_1 \mathcal{L}_2 = \sum_{\text{m-type } \mathcal{L}_{i_m}} \dim_{\mathbb{C}} (V_{\mathcal{L}_1, \mathcal{L}_2, \mathcal{L}_{i_m}}) \mathcal{L}_{i_m} + \sum_{\text{q-type } \mathcal{L}_{i_q}} \dim_{\mathbb{C}^{1|1}} (V_{\mathcal{L}_1, \mathcal{L}_2, \mathcal{L}_{i_q}}) \mathcal{L}_{i_q} \, . \tag{2.42}$$

The derivation of this formula is given in Appendix C. Due to this formula, the dimension of the junction vector space will be also referred as the multiplicity.

Since the TDL $\mathcal{L}_{i_q}$ is q-type, the dimensions of the bosonic and fermionic subspaces of $V_{\mathcal{L}_1, \mathcal{L}_2, \mathcal{L}_{i_q}}$ coincide, i.e.

$$\dim_{\mathbb{C}^{1|1}} (V_{\mathcal{L}_1, \mathcal{L}_2, \mathcal{L}_{i_q}}) = \dim_{\mathbb{C}} (V^{\text{f}}_{\mathcal{L}_1, \mathcal{L}_2, \mathcal{L}_{i_q}}) = \dim_{\mathbb{C}} (V^{\text{f}}_{\mathcal{L}_1, \mathcal{L}_2, \mathcal{L}_{i_q}}) \equiv n_{12 i_q} \, . \tag{2.43}$$

On the other hand, the dimensions of the bosonic and fermionic subspaces of $V_{\mathcal{L}_1, \mathcal{L}_2, \mathcal{L}_{i_m}}$ are different in general, and we define

$$n_{\text{b}, 12 i_m} \equiv \dim_{\mathbb{C}} (V^{\text{b}}_{\mathcal{L}_1, \mathcal{L}_2, \mathcal{L}_{i_m}}) \, , \quad n_{\text{f}, 12 i_m} \equiv \dim_{\mathbb{C}} (V^{\text{f}}_{\mathcal{L}_1, \mathcal{L}_2, \mathcal{L}_{i_m}}) \, . \tag{2.44}$$

---

[16]We thank Ken Kikuchi for pointing this out to us.

For example, the fusion of a q-type TDL $\mathcal{L}$ with its orientation reversal gives

$$\mathcal{L}\overline{\mathcal{L}} = \dim_{\mathbb{C}}(V_{\mathcal{L},\overline{\mathcal{L}},I})I + \cdots = (1_{\mathrm{b}} + 1_{\mathrm{f}})I + \cdots , \tag{2.45}$$

where we use $1_{\mathrm{b}}$ and $1_{\mathrm{f}}$ to emphasize the fact that $V^{\mathrm{b}}_{\mathcal{L},\overline{\mathcal{L}},I}$ and $V^{\mathrm{f}}_{\mathcal{L},\overline{\mathcal{L}},I}$ are both one-dimensional.

### 2.2.6   q-type invertible TDL and modified Cardy condition

Let us consider a set of $N_{\mathrm{m}}$ m-type invertible TDLs $\mathcal{L}_{i_{\mathrm{m}}}$ for $i_{\mathrm{m}} = 1, \cdots , N_{\mathrm{m}}$ and $N_{\mathrm{q}}$ q-type TDLs $\mathcal{L}_{i_{\mathrm{q}}}$ for $i_{\mathrm{q}} = 1, \cdots , N_{\mathrm{q}}$, that closes under the fusion. Invertibility of the m-type TDLs implies

$$\mathcal{L}_{i_{\mathrm{m}}}\overline{\mathcal{L}}_{i_{\mathrm{m}}} = I . \tag{2.46}$$

We further impose the following fusion rules for the q-type TDLs

$$\mathcal{L}_{i_{\mathrm{q}}}\overline{\mathcal{L}}_{i_{\mathrm{q}}} = 2I . \tag{2.47}$$

A q-type TDL that satisfies the above fusion rule is called a *q-type invertible TDL*, by the reason we will see momentarily. The above fusion rules (2.46) and (2.47) imply that absolute values of the loop expectation values (quantum dimensions) are

$$|\langle \mathcal{L}_{i_m} \rangle_{\mathbb{R}^2}| = 1 , \quad |\langle \mathcal{L}_{i_q} \rangle_{\mathbb{R}^2}| = \sqrt{2} . \tag{2.48}$$

This further implies that the products $\mathcal{L}_{i_{\mathrm{m}}}\mathcal{L}_{j_{\mathrm{q}}}$ and $\mathcal{L}_{j_{\mathrm{q}}}\mathcal{L}_{i_{\mathrm{m}}}$ are q-type TDLs, the product $\mathcal{L}_{i_{\mathrm{m}}}\mathcal{L}_{j_{\mathrm{m}}}$ is a m-type TDL, and the product $\mathcal{L}_{i_{\mathrm{q}}}\mathcal{L}_{j_{\mathrm{q}}}$ is 2 times a m-type TDL, i.e. the fusion rules should take the form

$$\begin{aligned}
\mathcal{L}_{i_{\mathrm{m}}}\mathcal{L}_{j_{\mathrm{q}}} &= \mathcal{L}_{k_{\mathrm{q}}} , \quad \mathcal{L}_{j_{\mathrm{q}}}\mathcal{L}_{i_{\mathrm{m}}} = \mathcal{L}_{l_{\mathrm{q}}} , \\
\mathcal{L}_{i_{\mathrm{m}}}\mathcal{L}_{j_{\mathrm{m}}} &= \mathcal{L}_{k_{\mathrm{m}}} , \quad \mathcal{L}_{i_{\mathrm{q}}}\mathcal{L}_{j_{\mathrm{q}}} = 2\mathcal{L}_{k_{\mathrm{m}}} .
\end{aligned} \tag{2.49}$$

Note that the above fusion rules admit an extra $\mathbb{Z}_2$ grading, where the m-type TDLs are $\mathbb{Z}_2$ even and the q-type TDLS are $\mathbb{Z}_2$ odd, but one should not confused this with the $\mathbb{Z}_2$ grading $\sigma$ of the defect Hilbert vector space.

The above fusions are almost group multiplications, except that the coefficient on the RHS of the last equation in (2.49) is not 1. If we define

$$\widetilde{\mathcal{L}}_{i_q} = \frac{1}{\sqrt{2}}\mathcal{L}_{i_q} \tag{2.50}$$

and ignore the differences between $1_{\mathrm{b}}$ and $1_{\mathrm{f}}$, then the fusion forms a group with the elements $\{\mathcal{L}_{i_m}, \widetilde{\mathcal{L}}_{i_q}\}$.[17]   The TDLs $\widetilde{\mathcal{L}}_{i_q}$ together with the m-type invertible TDLs $\mathcal{L}_{i_m}$ generate an

---

[17]Dividing a factor of $\sqrt{2}$ in (2.50) could be interpreted as factoring out the 1d Majorana fermion living on the worldline of the TDLs $\mathcal{L}_{i_{\mathrm{q}}}$.

(invertible) symmetry by their actions on local operators and states in the fermionic CFT.[18] Note importantly that some parts of the symmetry are generated by $\widetilde{\mathcal{L}}_{i_q}$ that do not have a well-defined defect Hilbert space. Hence, the usual Cardy condition should be modified. Consider the twisted partition function

$$Z^{\widetilde{\mathcal{L}}_{i_q}}(\tau, \bar{\tau}) = \mathrm{Tr}_{\mathcal{H}}\left(\widehat{\widetilde{\mathcal{L}}}_{i_q} q^{L_0 - \frac{c}{24}} \bar{q}^{\widetilde{L}_0 - \frac{c}{24}}\right). \tag{2.51}$$

Its S-transformation equals to $\frac{1}{\sqrt{2}}$ times the partition function for the defect Hilbert space $\mathcal{H}_{\mathcal{L}_{i_q}}$

$$Z^{\widetilde{\mathcal{L}}_{i_q}}\left(-\frac{1}{\tau}, -\frac{1}{\bar{\tau}}\right) = \frac{1}{\sqrt{2}} \mathrm{Tr}_{\mathcal{H}_{\mathcal{L}_{i_q}}}\left(q^{L_0 - \frac{c}{24}} \bar{q}^{\widetilde{L}_0 - \frac{c}{24}}\right). \tag{2.52}$$

The spectrum of the defect Hilbert space $\mathcal{H}_{\mathcal{L}_{i_q}}$ is doubly degenerate because $\mathcal{H}_{\mathcal{L}_{i_q}}$ forms a representation of a one-dimensional Clifford algebra generated by the action of the 1d Majorana fermion $\psi$ living on the q-type TDL $\mathcal{L}_{i_q}$. Therefore, $Z^{\widetilde{\mathcal{L}}_{i_q}}\left(-\frac{1}{\tau}, -\frac{1}{\bar{\tau}}\right)$ should admit a $q$, $\bar{q}$-expansion with coefficients in $\sqrt{2}\mathbb{Z}_{\geq 0}$. This condition was referred to as the *modified Cardy condition* in [16].

## 2.3   Relation to super fusion categories

The relation between TDLs in fermionic CFTs and super fusion categories is analogous to the relation between TDLs in bosonic CFTs and fusion categories discussed in [8]. Here, we highlight some modifications and additions, following the treatment of super fusion category in [72].

The defect Hilbert space $\mathcal{H}_{\mathcal{L}}$ of a TDL $\mathcal{L}$ is an object in the super fusion category. The morphisms between objects in the super fusion category are given by junction vectors in the two-way junction vector spaces. More explicitly, a weight $(h, \tilde{h}) = (0, 0)$ operator $m$ in the defect Hilbert space $\mathcal{H}_{\mathcal{L}_1, \overline{\mathcal{L}}_2}$ (i.e. $m \in V_{\mathcal{L}_1, \overline{\mathcal{L}}_2}$) gives an isomorphism between the defect Hilbert spaces,

$$m : \mathcal{H}_{\mathcal{L}_1} \to \mathcal{H}_{\mathcal{L}_2}. \tag{2.53}$$

The endomorphism spaces for m-type and q-type TDLs are

$$\mathrm{End}(\mathcal{H}_{\mathcal{L}_\mathrm{m}}) = V_{\mathcal{L}_\mathrm{m}, \overline{\mathcal{L}}_\mathrm{m}} \cong \mathbb{C}, \quad \mathrm{End}(\mathcal{H}_{\mathcal{L}_\mathrm{q}}) = V_{\mathcal{L}_\mathrm{q}, \overline{\mathcal{L}}_\mathrm{q}} \cong \mathbb{C}^{1|1}. \tag{2.54}$$

The junction vector space $V_{\mathcal{L}_{i_1}, \cdots, \mathcal{L}_{i_n}}$ (the subspace of the weight-$(0, 0)$ operators in $\mathcal{H}_{\mathcal{L}_{i_1}, \cdots, \mathcal{L}_{i_n}}$) corresponds to the fusion space $V_{i_n}^{i_{n-1}, \cdots, i_1}$ in the super fusion category [72]. In the discussion of the F-moves, the tensor products between the fusion spaces are usually taken to have

---

coefficients valued in the endomorphism spaces (2.54) (as in (351) of [72]). It is equivalent to considering the projected tensor products (2.14) with $\mathbb{C}$-number coefficients. An important new ingredient of the pentagon equation in the super fusion category is the Koszul sign from exchanging the order of the fusion spaces. Such a sign corresponds to the sign (2.21) from exchanging the ordering of the junction vectors.

[18]We adopt the strict sense of symmetry here, as opposed to the non-invertible or category symmetry.

In super fusion categories, there exist an object $\Pi$ called "transparent fermion" with the distinguishing property [72, 13][19]

$$\mathrm{Hom}(\Pi, I) = \mathbb{C}^{0|1} . \tag{2.55}$$

However, we would not regard this object as corresponding to a genuine nontrivial TDL because of the following reasons. By (2.55), the defect Hilbert space of the transparent fermion TDL $\Pi$ contains a fermionic topological operator $\psi$ (with conformal dimension $(h, \tilde{h}) = (0,0)$).[20] By pair-creating and pair-annihilating $\psi$, a TDL $\Pi$ can break and shrink to disappear or to points on q-type TDLs. Therefore, any TDL configuration involving the TDL $\Pi$ reduces to the configuration without $\Pi$. For example, consider a TDL $\Pi$ wrapping a local operator $\mathcal{O}$. The $\Pi$ line can unwrap by pair-creating $\psi$'s and shrink to disappear by pair-annihilating the $\psi$'s,

$$\widehat{\Pi}(\mathcal{O}) = \quad \text{} \quad = \quad \text{} \quad = \mathcal{O} . \tag{2.56}$$

Another example is a TDL $\Pi$ stretching between a fermionic topological operator $\psi$ and a q-type TDL $\mathcal{L}_\mathrm{q}$. The $\Pi$ line can shrink to a point on $\mathcal{L}_\mathrm{q}$, and $\psi$ becomes the 1d Majorana fermion on $\mathcal{L}_\mathrm{q}$,

$$\text{} \tag{2.57}$$

---

[19]A super-fusion category $\mathcal{C}$ containing a $\Pi$ line is known as a $\Pi$-superfusion category [85, 86], which is equipped with a distinguished object $\Pi$ and every other object $\mathcal{L}$ in $\mathcal{C}$ is oddly isomorphic to $\Pi\mathcal{L}$.

[20]Here, we abuse the notation a bit to denote the TDL corresponding to the object $\Pi$ also by $\Pi$.

# 3 Super fusion categories of rank-2

Let us consider a system of two simple TDLs, the trivial TDL $I$ and a non-trivial TDL $W$. The system is described by super fusion categories of rank-2. Let us first consider the case when $W$ is m-type. The consistency of the F-moves with two external $I$'s and two external $W$'s implies that the one-dimensional junction vector spaces $V_{W,I,W}$ and $V_{I,W,W}$ should be bosonic while $V_{W,W,I}$ can be bosonic or fermionic. Hence, the most general fusion rules are

$$W^2 = 1_{\mathrm{b}} I + (n_{\mathrm{b}} + n_{\mathrm{f}}) W , \quad WI = 1_{\mathrm{b}} W = IW , \tag{3.1}$$

or

$$W^2 = 1_{\mathrm{f}} I + (n_{\mathrm{b}} + n_{\mathrm{f}}) W , \quad WI = 1_{\mathrm{b}} W = IW , \tag{3.2}$$

where we added subscripts b and f to the fusion numbers to distinguish the dimensionalities of the bosonic and fermionic junction vector spaces. The corresponding (super-)fusion categories are labeled as $\mathcal{C}_{\mathrm{m}}^{(n_{\mathrm{b}},n_{\mathrm{f}})}$ and $\widehat{\mathcal{C}}_{\mathrm{m}}^{(n_{\mathrm{b}},n_{\mathrm{f}})}$. Next, let us consider the case when $W$ is $q$-type. The junction vector spaces $V_{W,I,W}$, $V_{I,W,W}$, and $V_{W,W,I}$ are all isomorphic to $\mathbb{C}^{1|1}$. Hence, the most general fusion rule is

$$W^2 = (1_{\mathrm{b}} + 1_{\mathrm{f}}) I + n W , \quad WI = W = IW , \tag{3.3}$$

The corresponding super fusion categories are labeled as $\mathcal{C}_{\mathrm{q}}^n$.

We solve the super pentagon identity (2.20) for $(n_{\mathrm{b}}, n_{\mathrm{f}}) \in \{(0,0), (1,0), (0,1), (2,0), (1,1),$ $(0,2), (3,0), (2,1), (1,2), (0,3), (3,1), (2,2), (1,3)\}$, and $n \leq 2$, in the following procedure. We first use `Mathematica` to spell out the super pentagon identity as polynomial equations whose variables are the F-symbols. We then use `Magma` to find the Gröbner basis of the polynomial equations. Finally, we input the Gröbner basis back to `Mathematica` to solve for the solutions. We find that the solutions only exist for $\mathcal{C}_{\mathrm{m}}^{(0,0)}$, $\mathcal{C}_{\mathrm{m}}^{(1,0)}$, $\mathcal{C}_{\mathrm{m}}^{(0,1)}$, $\mathcal{C}_{\mathrm{m}}^{(1,1)}$, $\widehat{\mathcal{C}}_{\mathrm{m}}^{(0,0)}$, $\mathcal{C}_{\mathrm{q}}^0$, and $\mathcal{C}_{\mathrm{q}}^2$. We conjecture that these are all the rank-2 super fusion categories.

The $\mathcal{C}_{\mathrm{m}}^{(0,0)}$, $\widehat{\mathcal{C}}_{\mathrm{m}}^{(0,0)}$, and $\mathcal{C}_{\mathrm{q}}^0$ categories correspond to $\mathbb{Z}_2$ symmetry, i.e. their fusion rings (after the proper rescaling in Section 2.2.6) are $\mathbb{Z}_2$ group ring. They fall into a $\mathbb{Z}_8$ classification [73–77] and can be realized by the tensor product of $\nu$ copies of $m = 3$ fermionic minimal models (free Majorana fermions) [87–89, 73], as summarized in Table 1.[21] We comment here that the $\widehat{\mathcal{C}}_{\mathrm{m}}^{(0,0)}$ super fusion categories have some very unpleasant features that seem to violate the locality property reviewed in Section 1.1. We will discuss this problem and a potential resolution in Section 3.2.4.

---

[21]We thank the JHEP referee for emphasizing the $\mathbb{Z}_8$ classification.

| | | # categories | $\nu$ |
|---|---|---|---|
| $\mathcal{C}_{\mathrm{m}}^{(0,0)}$ | 2 | non-anomalous | 0 mod 8 |
| | | anomalous | 4 mod 8 |
| $\widehat{\mathcal{C}}_{\mathrm{m}}^{(0,0)}$ | 2 | $\kappa = -1$ | 2 mod 8 |
| | | $\kappa = 1$ | 6 mod 8 |
| $\mathcal{C}_{\mathrm{q}}^{0}$ | 4 | $\kappa = 1,\ \gamma = e^{\frac{3i\pi}{4}}$ | 1 mod 8 |
| | | $\kappa = -1,\ \gamma = e^{\frac{i\pi}{4}}$ | 3 mod 8 |
| | | $\kappa = -1,\ \gamma = e^{\frac{3i\pi}{4}}$ | 5 mod 8 |
| | | $\kappa = 1,\ \gamma = e^{\frac{i\pi}{4}}$ | 7 mod 8 |

Table 1: The rank-2 super fusion categories of $\mathbb{Z}_2$ symmetries and their fermionic CFT realizations.

The other categories $\mathcal{C}_{\mathrm{m}}^{(1,0)}$, $\mathcal{C}_{\mathrm{m}}^{(0,1)}$, $\mathcal{C}_{\mathrm{m}}^{(1,1)}$, and $\mathcal{C}_{\mathrm{q}}^{2}$ correspond to non-invertible symmetries, i.e. their fusion rings are not group rings. We list the number of categories for each case and their CFT realizations in the table below.

| | # categories with $\langle W \rangle_{\mathbb{R}^2} > 0$ | # categories with $\langle W \rangle_{\mathbb{R}^2} < 0$ | CFT realizations with $\langle W \rangle_{\mathbb{R}^2} > 0$ |
|---|---|---|---|
| $\mathcal{C}_{\mathrm{m}}^{(1,0)}$ | 1 | 1 | $m = 4$ fermionic minimal model |
| $\mathcal{C}_{\mathrm{m}}^{(0,1)}$ | 1 | 1 | ? |
| $\mathcal{C}_{\mathrm{m}}^{(1,1)}$ | 1 | 1 | $m = 7$ fermionic minimal model |
| $\mathcal{C}_{\mathrm{q}}^{2}$ | 2 | 2 | exceptional $m = 11$ fermionic minimal model |
| | | | exceptional $m = 12$ fermionic minimal model |

Table 2: The rank-2 super fusion categories of non-invertible symmetries and the fermionic CFT realizations, of those with a positive loop expectation value $\langle W \rangle_{\mathbb{R}^2} > 0$, in the fermionic minimal models [68, 69, 79]. We leave a question mark for the $\mathcal{C}_{\mathrm{m}}^{(0,1)}$ categories, as we do not know their CFT realization.

## 3.1 Gauge fixings

Before proceeding to solve the super pentagon identity, we first briefly discuss some generalities on the gauge fixings. As we have seen in Section 2.1, for every trivalent junction, one is free to change the basis of the junction vector space. When the junction vector space is

one-dimensional, this corresponds to a simple rescaling of the graph by a complex number,

$$
\begin{array}{c}
a \diagdown \quad \diagup b \\
\phantom{a} \alpha \\
| \\
c
\end{array}
\quad \rightarrow \quad
g_\alpha^{a,b,c}
\begin{array}{c}
a \diagdown \quad \diagup b \\
\phantom{a} \alpha \\
| \\
c
\end{array}
\tag{3.4}
$$

where $a$, $b$, $c$, label the type of TDLs and $\alpha = 0$ or $1$ for the bosonic or fermionic junction. If the junction vector space is multi-dimensional, say

$$
V_{a,b,c} = \mathbb{C}^{m|n}, \tag{3.5}
$$

the above scaling factor will be promoted to a $m \times m$ or $n \times n$ matrices for $\alpha = 0$ or $1$, i.e.

$$
\begin{array}{c}
a \diagdown \quad \diagup b \\
\phantom{a} \alpha, i \\
| \\
c
\end{array}
\quad \rightarrow \quad
\left( g_\alpha^{a,b,c} \right)_j^i
\begin{array}{c}
a \diagdown \quad \diagup b \\
\phantom{a} \alpha, j \\
| \\
c
\end{array}
\tag{3.6}
$$

where the addition indexes $i$, $j$ label the multiplicities. Correspondingly, the F-matrices for the following graph

$$
\begin{array}{c}
a \quad b \quad c \\
\alpha, i \diagup e \\
\beta, j | \\
d
\end{array}
\quad = \quad
\sum_{f,\delta,\gamma} \sum_{k,l}
\begin{array}{c}
a \quad b \quad c \\
f \diagup \gamma, k \\
\delta, l | \\
d
\end{array}
\mathcal{F}_{d;\,\delta,\gamma}^{a,b,c;\,\alpha,\beta}(e,f)_{k,l}^{i,j}, \tag{3.7}
$$

will transform as

$$
\mathcal{F}_{d;\,\delta,\gamma}^{a,b,c;\,\alpha,\beta}(e,f)_{k,l}^{i,j} \longrightarrow \sum_{i',j',k',l'} \left( g_\gamma^{b,c,f} \right)_k^{k'} \cdot \left( g_\delta^{a,f,d} \right)_l^{l'} \cdot \left( g_\alpha^{a,b,e} \right)_{i'}^{-1\,i} \cdot \left( g_\beta^{e,c,d} \right)_{j'}^{-1\,j} \mathcal{F}_{d;\,\delta,\gamma}^{a,b,c;\,\alpha,\beta}(e,f)_{k',l'}^{i',j'}.
$$

$$\tag{3.8}$$

In section 3.3.2, we will solve the super pentagon identities of the $\mathcal{C}_q^2$ categories, which have multiplicity two for both the bosonic and fermionic fusion channels. For their gauge fixings, we will present a detailed discussion in App.D. In the discussion below, for simplicity, we focus on the case of multiplicity one. In this case, the F-matrices transform as

$$
\mathcal{F}_{d;\,\delta,\gamma}^{a,b,c;\,\alpha,\beta}(e,f)_{k,l}^{i,j} \longrightarrow \frac{g_\gamma^{b,c,f} \cdot g_\delta^{a,f,d}}{g_\alpha^{a,b,e} \cdot g_\beta^{e,c,d}} \mathcal{F}_{d;\,\delta,\gamma}^{a,b,c;\,\alpha,\beta}(e,f). \tag{3.9}
$$

It is easy to verify that the super pentagon equations (2.20) are homogeneous under the scaling (3.9). Therefore, we need to first choose some gauge conditions to remove the scaling

redundancies, which in turn helps us to fix some components of the F-matrices to certain values. Clearly, if two components of the F-matrices share the same scaling factor, one can only use the factor to fix one of the components, while the other one would be later determined by the super pentagon identities. Given a set of trivalent junctions and the F-matrices associated with them, we need to determine a maximal set of components of the F-matrices that can be gauge fixed. The gauge transformation (3.9) can be schematically written as

$$\mathcal{F}_\mu \longrightarrow \left( \prod_i^n g_i^{\mu_i} \right) \mathcal{F}_\mu \,, \tag{3.10}$$

where $i$ denotes the collection of the indices $(a, b, c; \alpha)$, and $\mu$ collectively denotes the indices of the F-matrices. The exponent $\mu_i$ can only take values in $\{-2,\ -1,\ 0,\ 1,\ 2\}$. Therefore, for each F-matrix $\mathcal{F}_\mu$, we can assign a "gauge vector" $\vec{v}$ to it,

$$\vec{v}_\mu = (\mu_1,\ \mu_2,\ \ldots,\ \mu_n)\,. \tag{3.11}$$

The gauge vectors for all the F-matrices form a matrix $\mathcal{G} = \{\vec{v}_\mu\}$. The maximal number of gauges we can choose is given by the rank of $\mathcal{G}$.

In practice, we can choose $\text{rank}(\mathcal{G})$ number of components of the F-matrices with linearly independent gauge vectors, and fix them to some convenient values. In the categories $\mathcal{C}_{\mathrm{m}}^{(1,0)}$ and $\mathcal{C}_{\mathrm{m}}^{(0,1)}$, there are three linearly independent gauge vectors, and thus we fix three components in the F-matrices. In $\mathcal{C}_{\mathrm{m}}^{(1,1)}$, there are four to fix. At last, in $\mathcal{C}_{\mathrm{q}}^0$, there are five to fix, which has been discussed throughout in Section 2.2.3, see (2.27) and (2.32). The overall scaling ($x_1 = x_2 = y_1 = y_2 = z_1 = z_2$ in (2.26)) actually does not change any F-matrix.

## 3.2   m-type:

Let us present the F-symbols of the super fusion categories $\mathcal{C}_{\mathrm{m}}^{(n_{\mathrm{b}}, n_{\mathrm{f}})}$.

### 3.2.1   $\mathcal{C}_{\mathrm{m}}^{(0,0)}$ and $\mathcal{C}_{\mathrm{m}}^{(1,0)}$

In these two cases, they are ordinary fusion categories with two objects, which have been studied in [8]. The fusion category $\mathcal{C}_{\mathrm{m}}^{(0,0)}$ has two solutions corresponding to the (non-)anomalous $\mathbb{Z}_2$ symmetry.

To compare with the fermionic $\mathcal{C}_{\mathrm{m}}^{(0,1)}$ later, we here highlight the non-trivial F-moves of

the H-junctions in the bosonic $\mathcal{C}_{\mathrm{m}}^{(1,0)}$,

$$\vee \;=\; \vee \;,\qquad \begin{pmatrix} \; \end{pmatrix} \;=\; \begin{pmatrix} \zeta^{-1} & 1 \\ \zeta^{-1} & -\zeta^{-1} \end{pmatrix}\cdot \begin{pmatrix} \; \end{pmatrix}, \tag{3.12}$$

where $\zeta = \frac{1\pm\sqrt{5}}{2}$.

### 3.2.2 $\mathcal{C}_{\mathrm{m}}^{(0,1)}$

Our first non-trivial super-fusion category with a m-type TDL has the fusion rule,

$$W^2 = I + 1_f\, W\,. \tag{3.13}$$

Besides the universal sector, we solved the rest of the super-pentagon equations induced by the above fusion rule. We found two consistent solutions and determines the F-moves in the following H-junctions,

$$\vee \;=\; -\vee \;,\qquad \begin{pmatrix} \; \end{pmatrix} \;=\; \begin{pmatrix} \zeta^{-1} & 1 \\ -\zeta^{-1} & \zeta^{-1} \end{pmatrix}\cdot \begin{pmatrix} \; \end{pmatrix}, \tag{3.14}$$

where the dots at the vertices denote the fermionic junction vector in the junction vector space. In addition, we also used the gauge freedom,

$$\vee \;\rightarrow\; z\;\vee \;, \tag{3.15}$$

to normalize the upper right entry $\mathcal{F}_{W}^{W,W,W}(I,W)$ to 1.

### 3.2.3 $\mathcal{C}_{\mathrm{m}}^{(1,1)}$

In a similar fashion, for the super-category $\mathcal{C}_{\mathrm{m}}^{(1,1)}$ with fusion rule,

$$W^2 = I + (1_{\mathrm{b}} + 1_{\mathrm{f}})\, W\,, \tag{3.16}$$

we find two consistent solutions to the super pentagon equations. Besides the universal sector, we spell out these F-moves as follows:

$$\left( \begin{array}{c} \text{diagram} \end{array} \right) = \begin{pmatrix} 1 & 0 \\ 0 & -1 \end{pmatrix} \cdot \left( \begin{array}{c} \text{diagram} \end{array} \right),$$

$$\left( \begin{array}{c} \text{diagram} \end{array} \right) = \begin{pmatrix} \frac{1}{w} & -\frac{w}{w+1} & 1 & 0 & 0 \\ \frac{1-w}{w} & \frac{1}{w+1} & 1 & 0 & 0 \\ -\frac{1}{w} & -\frac{1}{2} & \frac{1-w}{2w} & 0 & 0 \\ 0 & 0 & 0 & -\frac{w}{w+1} & -\frac{w}{w+1} \\ 0 & 0 & 0 & -\frac{w}{w+1} & \frac{w}{w+1} \end{pmatrix} \cdot \left( \begin{array}{c} \text{diagram} \end{array} \right), \tag{3.17}$$

where $w = 1 \pm \sqrt{2}$ and we have once again used up the gauge freedoms,

$$\left( \begin{array}{c} \text{diagram} \end{array} \right) \rightarrow g_0 \left( \begin{array}{c} \text{diagram} \end{array} \right), \qquad \left( \begin{array}{c} \text{diagram} \end{array} \right) \rightarrow g_1 \left( \begin{array}{c} \text{diagram} \end{array} \right), \tag{3.18}$$

to normalize the 13 and 23 entries of $\mathcal{F}_W^{W,W,W}$ to 1. Using the above F-moves, we compute the loop expectation value

$$\langle W \rangle_{\mathbb{R}^2} = w. \tag{3.19}$$

We would like to further remark that the first solution is gauge equivalent to the F-moves in the super fusion category obtained by the fermion condensation of the $R_{\mathbb{C}}(\widehat{so(3)}_6)$ fusion category given in [78].

### 3.2.4 $\widehat{\mathcal{C}}_{\mathrm{m}}^{(0,0)}$

$\widehat{\mathcal{C}}_{\mathrm{m}}^{(0,0)}$ is an variant of the $\mathcal{C}_{\mathrm{m}}^{(0,0)}$ category with the trivalent junctions

$$\left( \begin{array}{c} \text{diagram} \end{array} \right), \qquad \left( \begin{array}{c} \text{diagram} \end{array} \right), \qquad \left( \begin{array}{c} \text{diagram} \end{array} \right), \tag{3.20}$$

where the first junction is fermionic and the other two are bosonic. By solving super-pentagon equations, we have the following non-trivial F-symbol,

$$= \quad \kappa i \quad \qquad , \qquad (3.21)$$

where $\kappa = \pm 1$.

While the above solutions give consistent super fusion categories, the trivalent junctions (3.20) are inconsistent with the locality property reviewed in Section 1.1, which implies the isomorphisms between the junction vector spaces $V_{W,W,I}$, $V_{I,W,W}$, $V_{W,I,W}$, and $V_{W,W}$. More explicitly, we can cut off a disc that contains one of the trivalent junctions in (3.20). No matter which junction is inside the disc, we have the same Hilbert space associated with the boundary circle, from quantization on the boundary circle. The Hilbert space has a unique ground state (with conformal weight $(h, \tilde{h}) = (0, 0)$). The fermion parity of the ground state should agree with the fermion parity of the junction inside the disc. However, the three junctions in (3.20) do not have the same fermion parity.

A potential resolution to this problem is that we demand the fermionic junctions on single TDL $W$ must always come in pairs, which cannot be separated by cutting off or gluing back discs. In particular, the number of fermionic junctions inside a disc should always be even. With this constraint, the locality property would only imply isomorphisms between the tensor product space $V_{W,W,I} \otimes V_{W,W,I}$ and the spaces $V_{I,W,W}$, $V_{W,I,W}$, and $V_{W,W}$. On the other hand, this constraint would not affect the derivation of the F-moves in Section 2.2.1, since we could always add an extra fermionic junction on one of the nontrivial external lines.

## 3.3 q-type:

We present the F-symbols of the super fusion categories $\mathcal{C}_q^n$.

### 3.3.1 $\mathcal{C}_q^0$

$\mathcal{C}_q^0$ is the first non-trivial super-fusion category of rank-2 containing a q-type TDL, with the following fusion rule,

$$W^2 = (1_b + 1_f)I. \qquad (3.22)$$

The q-type TDL is always unoriented for the category of rank-2. In Section 2.2.3, we have solved the F-matrices $\mathcal{F}_I^{W,W,I}$, $\mathcal{F}_I^{W,I,W}$, $\mathcal{F}_W^{W,I,I}$, $\mathcal{F}_I^{I,W,W}$, $\mathcal{F}_W^{I,W,I}$ and $\mathcal{F}_W^{I,I,W}$. There are two

gauge inequivalent solutions given in (2.33). With respect to each of the above solutions, one can further find two inequivalent solutions to the F-matrix $\mathcal{F}_W^{W,W,W}$. So overall there are *four* inequivalent solutions. We list them as follows:

$$
\begin{pmatrix} \\ \\ \\ \\ \end{pmatrix} = \frac{\kappa}{\sqrt{2}} \begin{pmatrix} 1 & \gamma & 0 & 0 \\ \gamma & -\gamma^2 & 0 & 0 \\ 0 & 0 & 1 & \gamma \\ 0 & 0 & \gamma & -\gamma^2 \end{pmatrix} \cdot \begin{pmatrix} \\ \\ \\ \\ \end{pmatrix} , \tag{3.23}
$$

for $\kappa = \pm 1$ and $\gamma = e^{\frac{i\pi}{4}}, e^{\frac{3i\pi}{4}}$.[22] Using the above F-moves, we compute the loop expectation value

$$
\langle W \rangle_{\mathbb{R}^2} = \sqrt{2}\kappa . \tag{3.24}
$$

The solution with $\kappa = 1$ and $\gamma = e^{\frac{3\pi i}{4}}$ is gauge equivalent to the F-moves in the super fusion category obtained by the fermion condensation of the $\mathbb{Z}_2$ TY fusion category given in [78]. Actually, all of the four solutions can be obtained via the fermionic anyon condensation of the 3d Ising TQFT. In the condensation picture, there are two ways [91] to braid the anyons $\sigma$ and $\psi$, corresponding to the duality $\mathcal{N}$-line and the $\mathbb{Z}_2$ $\eta$-line in the 2d Ising CFT. The two braidings are precisely encoded in the values of $\gamma$, while the Frobenius-Schur indicator $\kappa$ is inherited from its bosonic parent. We thus have overall four ways of fermionic condensations.[23]

### 3.3.2 $\mathcal{C}_q^2$

Our last example is the $\mathcal{C}_q^2$ super fusion categories, whose fusion rule is given by

$$
W^2 = (1_b + 1_f)I + 2W , \tag{3.25}
$$

---

[22]The F-moves of the $\mathcal{C}_q^0$ categories are also obtained in [90]. We thank Yanzhen Wang for sharing his undergraduate thesis with us.

[23]We thank Qing-Rui Wang for a discussion on this result from the perspective of fermionic condensation.

where $W$ is a q-type TDL. We separate the non-trivial F-symbols in the following four sectors classified by the number of their external trivial TDL:

$$
\begin{aligned}
\mathcal{S}_4 &: \quad \mathcal{F}_I^{I,I,I} \, ; \\
\mathcal{S}_2 &: \quad \mathcal{F}_I^{W,W,I}, \quad \mathcal{F}_I^{W,I,W}, \quad \mathcal{F}_W^{W,I,I}, \quad \mathcal{F}_I^{I,W,W}, \quad \mathcal{F}_W^{I,W,I}, \quad \text{and} \quad \mathcal{F}_W^{I,I,W} \, ; \\
\mathcal{S}_1 &: \quad \mathcal{F}_W^{I,W,W}, \quad \mathcal{F}_W^{W,I,W}, \quad \mathcal{F}_W^{W,W,I}, \quad \text{and} \quad \mathcal{F}_I^{W,W,W} \, ; \\
\mathcal{S}_0 &: \quad \mathcal{F}_W^{W,W,W} \, .
\end{aligned}
$$

Overall there are 361 variables in the F-symbols constrained by 9601 super-pentagons as well as the projection conditions. We find four solutions to these super-pentagons, where one of them is gauge equivalent to the solution obtained by fermionic condensation of the (bosonic) $\frac{1}{2}E_6$ fusion category as discussed in [78]. Now we spell them out in detail. One first needs to fix the gauge freedoms of these F-symbols and solve the projection conditions. We present the discussion of those in Appendix D. The trivial F-symbol $\mathcal{F}_I^{I,I,I} = 1$ in $\mathcal{S}_4$ as usual; The $\mathcal{S}_2$ sector is the universal sector for q-type TDLs that has been discussed before. We thus start from the sector $\mathcal{S}_1$. The F-matrices take the form (D.21) and (D.22). Using the solution (D.19),

$$
\sigma = \begin{pmatrix} 1 & 0 \\ 0 & 1 \end{pmatrix}, \quad \rho = \begin{pmatrix} 0 & 1 \\ -1 & 0 \end{pmatrix}, \quad \text{and} \quad \tau = \begin{pmatrix} 1 & 0 \\ 0 & -1 \end{pmatrix}, \tag{3.26}
$$

the super pentagon equations give

$$
f_{7b} = \frac{1}{2} \begin{pmatrix} 1 & 0 \\ 0 & 1 \end{pmatrix}, \quad \text{and} \quad f_{7f} = \frac{1}{2} \begin{pmatrix} 0 & 1 \\ -1 & 0 \end{pmatrix},
$$

$$
f_{8b} = \frac{1}{2} \begin{pmatrix} 1 & 0 \\ 0 & 1 \end{pmatrix}, \quad \text{and} \quad f_{8f} = \frac{1}{2} \begin{pmatrix} 0 & 1 \\ -1 & 0 \end{pmatrix},
$$

$$
f_{9b} = f_{9f} = \frac{1}{2} \begin{pmatrix} 1 & 0 \\ 0 & 1 \end{pmatrix},
$$

$$
f_{10b} = \frac{1}{8} \begin{pmatrix} \xi & -\frac{2}{\xi} \\ -\xi & -\frac{2}{\xi} \end{pmatrix} + \frac{\gamma^2}{8} \begin{pmatrix} \frac{2}{\xi} & \xi \\ -\frac{2}{\xi} & \xi \end{pmatrix}, \quad \text{and} \quad f_{10f} = \frac{1}{8} \begin{pmatrix} \frac{2}{\xi} & \xi \\ \frac{2}{\xi} & -\xi \end{pmatrix} + \frac{\gamma^2}{8} \begin{pmatrix} -\xi & \frac{2}{\xi} \\ -\xi & -\frac{2}{\xi} \end{pmatrix} .
$$

And finally, for the most non-trivial sector $\mathcal{S}_0$, the F-matrices take the form (D.24). Solving the super pentagon equations, we find the matrices $\mathcal{F}_b$ and $\mathcal{F}_f$ as,

$$
\mathcal{F}_b = \frac{1}{2}
\begin{pmatrix}
\frac{\xi}{2} & 0 & 0 & -\frac{1+\gamma^2}{2} & 2 & 2\gamma \\
0 & \frac{1+\gamma^2}{2} & \frac{\xi}{2} & 0 & -2\gamma^2 & -2\gamma^{-1} \\
-\frac{1+\gamma^2}{2} & 0 & 0 & -\frac{\xi}{2} & -2\gamma^2 & 2\gamma^{-1} \\
0 & -\frac{\xi}{2} & \frac{1+\gamma^2}{2} & 0 & 2 & -2\gamma \\
-\frac{1-\gamma^2}{4} - \frac{\xi}{4} & 0 & 0 & \frac{1+\gamma^2}{4} + \frac{\gamma^2 \xi}{4} & -\xi & -\gamma\,\xi \\
0 & \frac{\gamma+\gamma^{-1}}{4} + \frac{\gamma\xi}{4} & \frac{-\gamma+\gamma^{-1}}{4} + \frac{\gamma^{-1}\xi}{4} & 0 & -\gamma\,\xi & \gamma^2\xi
\end{pmatrix},
$$

$$
\mathcal{F}_f = \frac{1}{2}
\begin{pmatrix}
0 & \frac{1+\gamma^2}{2} & \frac{\xi}{2} & 0 & 2\gamma^{-1} & 2\gamma^2 \\
-\frac{\xi}{2} & 0 & 0 & \frac{1+\gamma^2}{2} & -2\gamma & -2 \\
0 & \frac{\xi}{2} & -\frac{1+\gamma^2}{2} & 0 & -2\gamma & 2 \\
-\frac{1+\gamma^2}{2} & 0 & 0 & -\frac{\xi}{2} & 2\gamma^{-1} & -2\gamma^2 \\
0 & -\frac{\gamma+\gamma^{-1}}{4} - \frac{\gamma\xi}{4} & \frac{\gamma-\gamma^{-1}}{4} - \frac{\gamma^{-1}\xi}{4} & 0 & \gamma^2\xi & -\gamma\,\xi \\
-\frac{1-\gamma^2}{4} - \frac{\xi}{4} & 0 & 0 & \frac{1+\gamma^2}{4} + \frac{\gamma^2\xi}{4} & -\gamma\,\xi & -\xi
\end{pmatrix},
$$

where we have defined $\xi = 1 \pm \sqrt{3}$, and $\gamma = e^{\frac{\pi i}{4}}$ or $e^{\frac{3\pi i}{4}}$ as before. Overall we have two choices of $\xi$'s and $\gamma$'s, and thus four solutions. From the F-moves, we compute the loop expectation value

$$
\langle W \rangle_{\mathbb{R}^2} = -\frac{2}{\xi}. \tag{3.27}
$$

Finally, the solution with $\xi = 1 - \sqrt{3}$ and $\gamma = e^{\frac{3\pi i}{4}}$ is gauge equivalent to the F-moves in the super fusion category obtained by the fermion condensation of the $\frac{1}{2}E_6$ fusion category given in [78].

# 4 Spin selection rules

Following the analysis in [8], the fractional part of the spins of the states in the bosonic or fermionic defect Hilbert space $\mathcal{H}_{\mathcal{L}}^{\mathrm{b}}$ or $\mathcal{H}_{\mathcal{L}}^{\mathrm{f}}$ can be determined by a sequence of modular T

transformations and the F-moves, schematically as

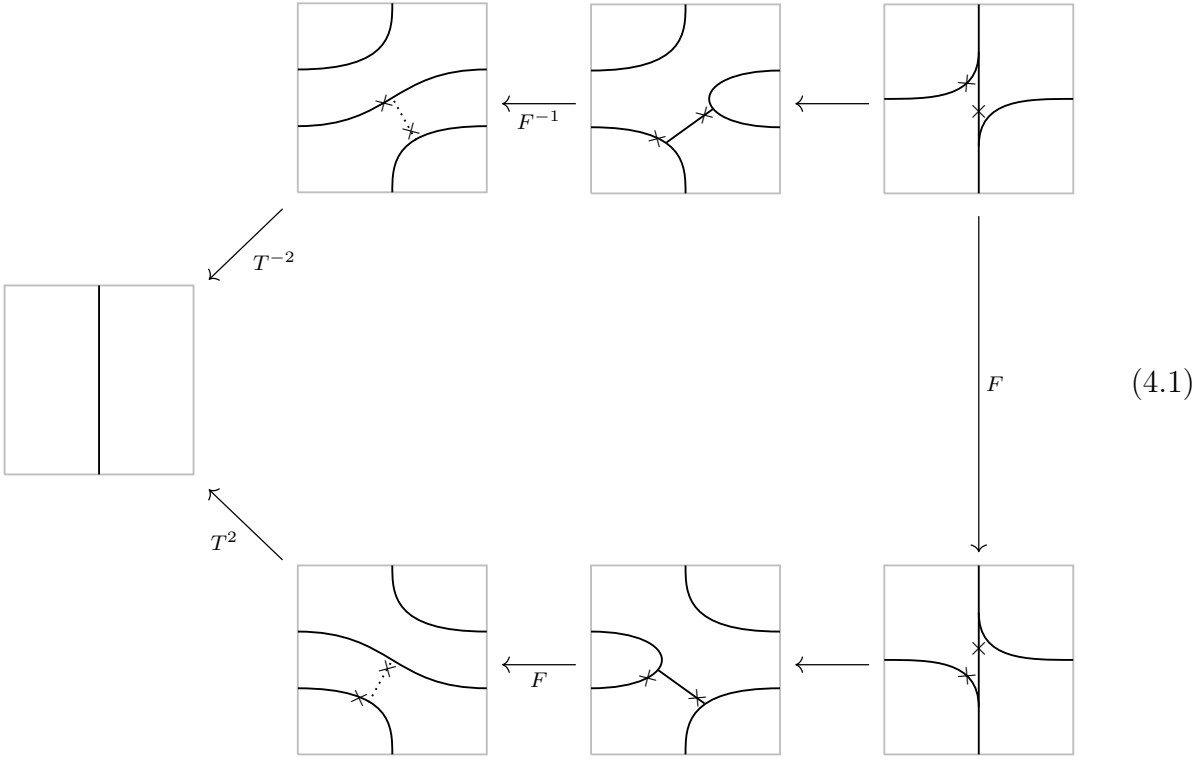

$$\text{(4.1)}$$

In the following subsections, we derive the spin selection rules for the super fusion categories $\mathcal{C}_{\mathrm{m}}^{(m,n)}$ and $\mathcal{C}_{\mathrm{q}}^{n}$. For simplicity, we will suppress the $\times$-marks and the trivial lines in the following TDL graphs, which could be recovered by comparing with the graphs in (4.1).

## 4.1  $\mathcal{C}_{\mathrm{m}}^{(1,0)}$

Let us first focus on the bosonic Fibonacci fusion category $\mathcal{C}_{\mathrm{m}}^{(1,0)}$ with the fusion rule

$$W^2 = I + W\,. \tag{4.2}$$

For any pair of states $|\psi\rangle, |\psi'\rangle \in \mathcal{H}_{\mathcal{L}}^{\mathrm{b}}$ or $\mathcal{H}_{\mathcal{L}}^{\mathrm{f}}$ with equal conformal weight, we consider the matrix element of the cylinder propagator

$$\langle \psi' | q^{L_0 - \frac{c}{24}} \bar{q}^{\widetilde{L}_0 - \frac{\tilde{c}}{24}} | \psi \rangle \quad = \quad \tag{4.3}$$

Performing modular $T^2$ transformation, and applying the F-move in (3.12), we find

$$\qquad = \quad \zeta^{-1} \quad \text{(diagram)} \quad + \quad \text{(diagram)} . \qquad (4.4)$$

Similarly, considering a modular $T^{-2}$ transformation on the matrix element $\langle \psi' | q^{L_0 - \frac{c}{24}} \bar{q}^{\widetilde{L}_0 - \frac{\tilde{c}}{24}} | \psi \rangle$ gives

$$\qquad = \quad \zeta^{-1} \quad \text{(diagram)} \quad + \quad \text{(diagram)} . \qquad (4.5)$$

Applying the F-move again on the last graph in above, we obtain

$$\qquad = \quad \zeta^{-1} \quad \text{(diagram)} \quad - \quad \zeta^{-1} \quad \text{(diagram)} . \qquad (4.6)$$

In summary, we find the following three equations

$$e^{4\pi i(s+\frac{1}{2}\sigma)} = \zeta^{-1} + \widehat{W}_+ , \quad e^{-4\pi i(s+\frac{1}{2}\sigma)} = \zeta^{-1} + \widehat{W}_- , \quad \widehat{W}_- = \zeta^{-1} e^{2\pi i(s+\frac{1}{2}\sigma)} - \zeta^{-1} \widehat{W}_+ . \quad (4.7)$$

where $s$ and $\sigma$ are the spin and the $\mathbb{Z}_2$-grading of the states $|\psi\rangle$ and $|\psi'\rangle$, and $\widehat{W}_\pm$ are the lassoing linear operators acting on $\mathcal{H}_W$ defined by an $W$ line wrapping the spatial circle splitting over the temporal $W$ line as in the last graph in (4.4) and (4.4), respectively.

The solution to (4.7) gives the spin selection rules: the states in the bosonic defect Hilbert space $\mathcal{H}_W^{\mathrm{b}}$ should have spins in

$$s \in \mathbb{Z} + \begin{cases} 0, \pm\frac{2}{5} & \text{for} \quad \zeta = \frac{1+\sqrt{5}}{2} , \\ 0, \pm\frac{1}{5} & \text{for} \quad \zeta = \frac{1-\sqrt{5}}{2} , \end{cases} \qquad (4.8)$$

and the states in the fermionic defect Hilbert space $\mathcal{H}_W^{\mathrm{f}}$ should have spins in

$$s \in \mathbb{Z} + \begin{cases} \frac{1}{2}, \pm\frac{1}{10} & \text{for} \quad \zeta = \frac{1+\sqrt{5}}{2} , \\ \frac{1}{2}, \pm\frac{3}{10} & \text{for} \quad \zeta = \frac{1-\sqrt{5}}{2} . \end{cases} \qquad (4.9)$$

We discuss a realization of these spin selection rules by the TDLs in the $m = 4$ fermionic minimal model in Section 5.1.3.

## 4.2 $\mathcal{C}_{\mathrm{m}}^{(0,1)}$

Let us carry out the same manipulation in the fermionic Fibonacci fusion category $\mathcal{C}_{\mathrm{m}}^{(0,1)}$.

We first perform modular $T^2$ transformation, upon which we apply for the F-move in (3.14), and we find

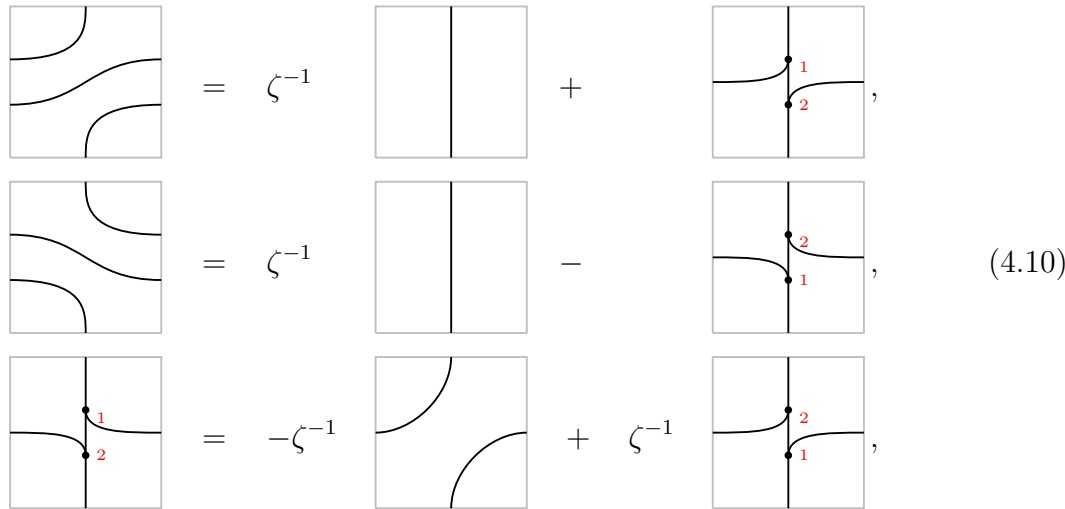

$$(4.10)$$

where the sign differences between (4.10) and (4.4), (4.6), (4.5) are due to the sign differences in the F-moves (3.14) and (3.12) and also minus signs coming from moving the fermionic junction vectors around the spatial circle. We find the equations

$$e^{4\pi i(s+\frac{1}{2}\sigma)} = \zeta^{-1} + \widehat{W}_+ , \quad e^{-4\pi i(s+\frac{1}{2}\sigma)} = \zeta^{-1} + \widehat{W}_- , \quad \widehat{W}_- = -\zeta^{-1}e^{2\pi i(s+\frac{1}{2}\sigma)} - \zeta^{-1}\widehat{W}_+ ,$$
$$(4.11)$$

which give the spin selection rules

$$s + \frac{\sigma}{2} \in \mathbb{Z} + \begin{cases} \frac{1}{2}, \pm\frac{1}{10} & \text{for} \quad \zeta = \frac{1+\sqrt{5}}{2} , \\ \frac{1}{2}, \pm\frac{3}{10} & \text{for} \quad \zeta = \frac{1-\sqrt{5}}{2} . \end{cases} \qquad (4.12)$$

Note interesting that the spin selection rules (4.12) are related to (4.8) and (4.9) with the fermion parity $(-1)^\sigma$ flipped.

## 4.3  $\mathcal{C}_{\mathrm{m}}^{(1,1)}$

Let us apply the manipulations in (4.1) to the super fusion category $\mathcal{C}_{\mathrm{m}}^{(1,1)}$:

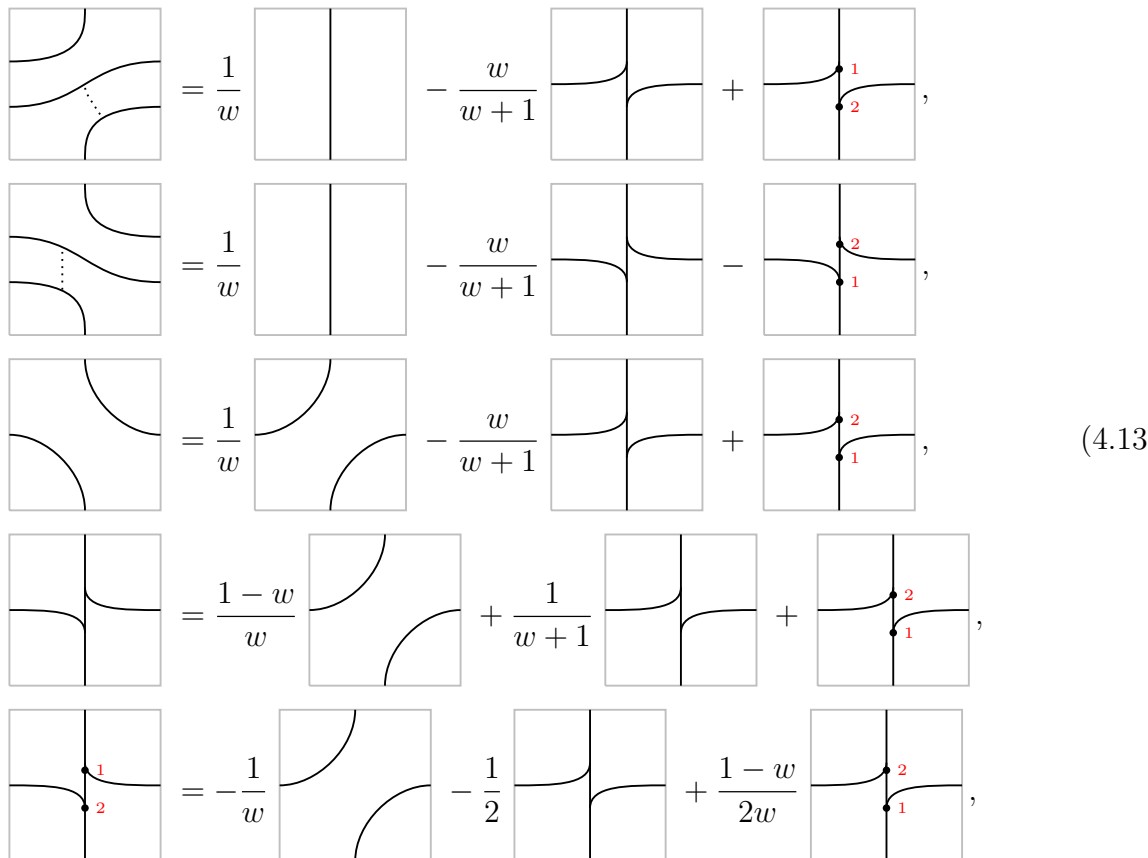

$$(4.13)$$

which gives the equations

$$e^{4\pi i(s+\frac{1}{2}\sigma)} = \frac{1}{w} - \frac{w}{w+1}\widehat{W}_+ - \widehat{W}_+^f\,,$$

$$e^{-4\pi i(s+\frac{1}{2}\sigma)} = \frac{1}{w} - \frac{w}{w+1}\widehat{W}_- + \widehat{W}_-^f\,,$$

$$e^{-2\pi i(s+\frac{1}{2}\sigma)} = \frac{1}{w} - \frac{w}{w+1}\widehat{W}_+ + \widehat{W}_+^f\,, \qquad (4.14)$$

$$\widehat{W}_- = \frac{1-w}{w}e^{2\pi i(s+\frac{1}{2}\sigma)} + \frac{1}{w+1}\widehat{W}_+ + \widehat{W}_+^f\,,$$

$$\widehat{W}_-^f = -\frac{1}{w}e^{2\pi i(s+\frac{1}{2}\sigma)} - \frac{1}{2}\widehat{W}_+ + \frac{1-w}{2w}\widehat{W}_+^f\,.$$

Solving the above equations, we find

$$s + \frac{\sigma}{2} \in \mathbb{Z} + \left\{0, \frac{1}{4}, \frac{1}{2}, \frac{3}{4}\right\} \quad \text{for} \quad w = 1 \pm \sqrt{2}\,. \qquad (4.15)$$

We discuss a realization of this spin selection rule by the TDLs in the $m = 7$ fermionic minimal model in Section 5.1.4.

## 4.4 $\widehat{\mathcal{C}}_{\mathrm{m}}^{(0,0)}$

Let us consider the super fusion category $\widehat{\mathcal{C}}_{\mathrm{m}}^{0,0}$ with the F-moves given in Section (3.2.4). Following the top sequence of the modular T transformations and the F-moves in (4.1), we find

$$
\begin{array}{ccccc}
\vcenter{\hbox{}} & = & \vcenter{\hbox{}} = \kappa i & \vcenter{\hbox{}} = \kappa i & \vcenter{\hbox{}}
\end{array}
, \tag{4.16}
$$

where we used the pair-creation and pair-annihilation of the fermionic junctions at the first and last equalities, respectively. We find the spin selection rules

$$
s = \frac{1}{2}\mathbb{Z} + \begin{cases} \frac{1}{8} & \text{for } \kappa = -1, \\ \frac{3}{8} & \text{for } \kappa = 1 . \end{cases} \tag{4.17}
$$

We discuss realizations of these spin selection rules by the TDLs in the tensor products of two or six copies of $m = 3$ fermionic minimal models (free Majorana fermions) in Section 5.1.2.

## 4.5 $\mathcal{C}_{\mathrm{q}}^0$

Let us consider the super fusion category $\mathcal{C}_{\mathrm{q}}^0$ with the F-moves given in Section 3.3.1. Following the bottom sequence of the modular T transformations and the F-moves in (4.1), we find

$$
\begin{aligned}
\vcenter{\hbox{}} &= \frac{\kappa}{\sqrt{2}} \vcenter{\hbox{}} + \frac{\kappa\gamma}{\sqrt{2}} \vcenter{\hbox{}} \\
&= \frac{\kappa}{\sqrt{2}} \vcenter{\hbox{}} - \frac{\kappa\gamma}{\sqrt{2}} \vcenter{\hbox{}} ,
\end{aligned} \tag{4.18}
$$

where again the black dots denote the 1d Majorana fermion on the q-type TDL. Next, we apply the rules in (2.25), and find

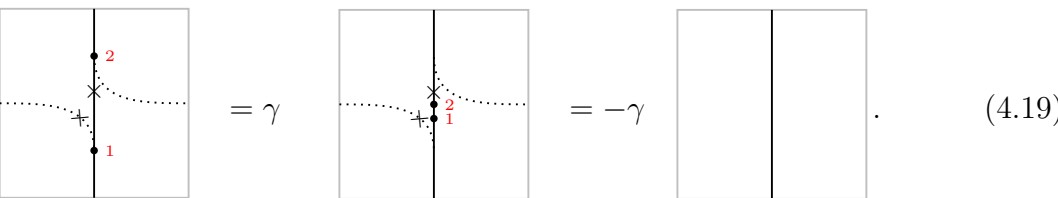

$$= \gamma \qquad\qquad = -\gamma \qquad\qquad . \qquad (4.19)$$

The spin in the defect Hilbert space $\mathcal{H}_W$ should satisfy the spin selection rules

$$s = \frac{1}{2}\mathbb{Z} + \begin{cases} \frac{7}{16} & \text{for } \kappa = 1, \ \gamma = e^{\frac{i\pi}{4}} , \\ \frac{3}{16} & \text{for } \kappa = -1, \ \gamma = e^{\frac{i\pi}{4}} , \\ \frac{1}{16} & \text{for } \kappa = 1, \ \gamma = e^{\frac{3i\pi}{4}} , \\ \frac{5}{16} & \text{for } \kappa = -1, \ \gamma = e^{\frac{3i\pi}{4}} . \end{cases} \qquad (4.20)$$

We discuss realizations of these spin selection rules by the TDLs in the $m = 3$ and $m = 4$ fermionic minimal models in Section 5.1.1 and 5.1.3.

## 4.6 $\mathcal{C}_{\mathrm{q}}^2$

To compute the spin selection rule for $\mathcal{C}_{\mathrm{q}}^2$, we need to use the bosonic part of the F-move (D.24), which is represented by the $10 \times 10$ matrix

$$\mathbf{F} \equiv \begin{pmatrix} 1 \otimes 1 & 0 \\ \tau^{-1} \otimes \rho^{-1} & 0 \\ 0 & 1 \end{pmatrix} \mathcal{F}_b \begin{pmatrix} 1 \otimes 1 & \tau \otimes \sigma & 0 \\ 0 & 0 & 1 \end{pmatrix} . \qquad (4.21)$$

The inverse of this F-move is represented by the pseudoinverse of $\mathbf{F}$,

$$\mathbf{F}^+ = \begin{pmatrix} \frac{1}{2}(1 \otimes 1) & 0 \\ \frac{1}{2}(\tau^{-1} \otimes \sigma^{-1}) & 0 \\ 0 & 1 \end{pmatrix} \mathcal{F}_b^{-1} \begin{pmatrix} \frac{1}{2}(1 \otimes 1) & \frac{1}{2}(\tau \otimes \rho) & 0 \\ 0 & 0 & 1 \end{pmatrix} . \qquad (4.22)$$

Let us define the following vectors

$$\mathbf{v}_1 = \left( \begin{array}{cccc} & & & \end{array} \right),$$

$$\mathbf{v}_2 = \left( \begin{array}{cccc} & & & \end{array} \right),$$

$$\mathbf{v}_3 = \left( \begin{array}{cccc} & & & \end{array} \right),$$

$$\mathbf{v}_4 = \left( \begin{array}{cccc} & & & \end{array} \right).$$
(4.23)

Applying the F-moves, we find the equations

$$\boxed{} = (\mathbf{F}^+ \mathbf{v}_1^T)_9, \qquad \boxed{} = (\mathbf{F} \mathbf{v}_2^T)_9, \qquad \mathbf{v}_4^T = \mathbf{F} \mathbf{v}_3^T.$$
(4.24)

Using the equations (2.25) and (D.1), we also find

$$\boxed{} = \gamma \boxed{}, \qquad \boxed{} = \gamma^{-1} \boxed{},$$

$$\boxed{} = \gamma \boxed{}, \qquad \boxed{} = \gamma^{-1} \boxed{},$$

$$\boxed{} = (\tau \otimes \sigma) \boxed{}, \qquad \boxed{} = (\tau \otimes \rho) \boxed{}.$$
(4.25)

Solving the equations (4.24) and (4.25), we find the spin selection rules

$$s + \frac{\sigma}{2} \in \mathbb{Z} + \begin{cases} 0, \tfrac{1}{6}, \tfrac{1}{2}, \tfrac{2}{3} & \text{for } \xi = 1 + \sqrt{3}, \ \gamma = e^{\frac{i\pi}{4}} \text{ or } \xi = 1 - \sqrt{3}, \ \gamma = e^{\frac{3i\pi}{4}}, \\ 0, \tfrac{1}{3}, \tfrac{1}{2}, \tfrac{5}{6} & \text{for } \xi = 1 + \sqrt{3}, \ \gamma = e^{\frac{3i\pi}{4}} \text{ or } \xi = 1 - \sqrt{3}, \ \gamma = e^{\frac{i\pi}{4}}. \end{cases}$$
(4.26)

We discuss realizations of these spin selection rules by the TDLs in the $m = 10$ and $m = 11$ exceptional fermionic minimal models in Section 5.2.

# 5  TDLs in the fermionic minimal models

In this section, we study TDLs in the fermionic minimal models [68,69,79]. We give a full set of simple TDLs in the standard models and a partial set of TDLs for the exceptional models. We further analyze the TDLS in the $m = 3, 4, 7$ standard models and the $m = 10, 11$ exceptional models, which give realizations for the rank-2 super fusion categories summarized in Table 2.

## 5.1  Standard models

The fermionic minimal models are labeled by an integer $m \in \mathbb{Z}_{\geq 3}$. For convenience, we introduce two integers $p$ and $q$ given by $p = m + 1$ and $q = m$ when $m$ is even, and $p = m$ and $q = m + 1$ when $m$ is odd. The holomorphic part of the primaries are labeled by $(r, s)$ with the range $1 \leq r \leq q - 1$ and $1 \leq s \leq \frac{p-1}{2}$ and holomorphic dimension $h_{r,s} = \frac{(pr-qs)^2 - 1}{4pq}$. The NS-sector partition function is [69]

$$Z_{NS} = \sum_{r \in [1]} \sum_{s=1}^{\frac{p-1}{2}} \chi_{r,s} \overline{\chi_{r,s}} + \sum_{r \in [\frac{q}{2}+1]} \sum_{s=1}^{\frac{p-1}{2}} \chi_{r,s} \overline{\chi_{q-r,s}} \,, \tag{5.1}$$

where $[n] = \{r | 1 \leq r \leq q - 1, \, r - n = 0 \mod 2\}$. We give the full set of TDLs in the NS-sector. There are q-type TDLs when $m = 0, 3 \mod 4$, and there is only m-type TDLs when $m = 1, 2 \mod 4$. We discuss these two cases separately.

First, for $m = 0, 3 \mod 4$, the m-type TDLs are

$$\mathcal{L}_{r,s}, \, (-1)^F \mathcal{L}_{r,s} \quad \text{for } r = 1, \cdots, \frac{q}{2} - 1 \text{ and } s = 1, \cdots, \frac{p-1}{2} \,, \tag{5.2}$$

and the q-type TDLs are

$$\mathcal{L}_{\frac{q}{2},s}, \, (-1)^F \mathcal{L}_{\frac{q}{2},s} \quad \text{for } s = 1, \cdots, \frac{p-1}{2} \,. \tag{5.3}$$

The NS-sector partition function twisted by the TDL $\mathcal{L}_{u,v}$ is

$$Z_{NS}^{\mathcal{L}_{u,v}} = \sum_{r \in [1]} \sum_{s=1}^{\frac{p-1}{2}} \frac{S_{(u,v),(r,s)}}{S_{(1,1),(r,s)}} \chi_{r,s} \overline{\chi_{r,s}} + \sum_{r \in [\frac{q}{2}+1]} \sum_{s=1}^{\frac{p-1}{2}} \frac{S_{(u,v),(r,s)}}{S_{(1,1),(r,s)}} \chi_{r,s} \overline{\chi_{q-r,s}} \,, \tag{5.4}$$

and the NS-sector partition function twisted by the TDL $(-1)^F \mathcal{L}_{u,v}$ is

$$Z_{NS}^{(-1)^F \mathcal{L}_{u,v}} = \sum_{r \in [1]} \sum_{s=1}^{\frac{p-1}{2}} \frac{S_{(u,v),(r,s)}}{S_{(1,1),(r,s)}} \chi_{r,s} \overline{\chi_{r,s}} - \sum_{r \in [\frac{q}{2}+1]} \sum_{s=1}^{\frac{p-1}{2}} \frac{S_{(u,v),(r,s)}}{S_{(1,1),(r,s)}} \chi_{r,s} \overline{\chi_{q-r,s}}, \tag{5.5}$$

where $S_{(u,v),(r,s)}$ is the modular S-matrix. It is straightforward to check the Cardy condition, i.e. the S-transformations of these twisted partition functions admit positive integer expansion in terms of the characters, and give the defect partition functions. The fusion ring is commutative with the fusion rules

$$((-1)^F)^2 = \mathcal{L}_{1,1} \equiv I,$$

$$\mathcal{L}_{r_1,s_1} \mathcal{L}_{r_2,s_2} = \sum_{r=1}^{\frac{q}{2}-1} \sum_{s=1}^{\frac{p-1}{2}} N_{(r_1,s_1),(r_2,s_2)}^{(r,s)} \mathcal{L}_{r,s} + \sum_{r=\frac{q}{2}}^{q-1} \sum_{s=1}^{\frac{p-1}{2}} N_{(r_1,s_1),(r_2,s_2)}^{(r,s)} \mathcal{L}_{q-r,s}, \tag{5.6}$$

where $N_{(r_1,s_1),(r_2,s_2)}^{(r,s)}$ is the fusion number which is related to the modular S-matrix by the Verlinde formula

$$N_{(r_1,s_1),(r_2,s_2)}^{(r,s)} = \sum_{u=1}^{q-1} \sum_{v=1}^{\frac{p-1}{2}} \frac{S_{(r_1,s_1),(u,v)} S_{(r_2,s_2),(u,v)} S_{(u,v),(r,s)}}{S_{(1,1),(u,v)}}. \tag{5.7}$$

Next, for $m = 1, 2 \mod 4$, the m-type TDLs are

$$\mathcal{L}_{r,s} \quad \text{for } r = 1, \cdots, q-1 \text{ and } s = 1, \cdots, \frac{p-1}{2}, \tag{5.8}$$

and there is no q-type TDL. The fermion parity TDL is given by $(-1)^F = \mathcal{L}_{q-1,1}$. The twisted partitions for the m-type TDLS are given by the same formula as (5.4). The fusion ring is commutative with the fusion rules

$$\mathcal{L}_{r_1,s_1} \mathcal{L}_{r_2,s_2} = \sum_{r=1}^{q-1} \sum_{s=1}^{\frac{p-1}{2}} N_{(r_1,s_1),(r_2,s_2)}^{(r,s)} \mathcal{L}_{r,s}. \tag{5.9}$$

Finally, it is not hard to explicitly check that (5.2) and (5.8) are all the simple TDLs for small $m$, and we conjecture that this holds for general $m$.

### 5.1.1  $m = 3$

In $m = 3$ fermionic minimal model, there are two m-type TDLs: $I \equiv \mathcal{L}_{1,1}$, $(-1)^F \equiv (-1)^F \mathcal{L}_{1,1}$, and two q-type TDLs: $\mathcal{L}_{2,1}$, $(-1)^F \mathcal{L}_{2,1}$. The fusion ring is commutative and subject to the relations

$$((-1)^F)^2 = I, \quad \mathcal{L}_{2,1}^2 = (1_b + 1_f)I. \tag{5.10}$$

The q-type TDLs $\mathcal{L}_{2,1}$ and $(-1)^F \mathcal{L}_{2,1}$ are invertible, and we define the rescaled TDLs

$$(-1)^{F_L} \equiv \frac{1}{\sqrt{2}} \mathcal{L}_{2,1}, \quad (-1)^{F_R} \equiv \frac{1}{\sqrt{2}} (-1)^F \mathcal{L}_{2,1}, \tag{5.11}$$

which generate the left and right fermion parities [16].

Taking the S-transformation of the twisted partition functions (5.4) and (5.5), we obtain the defect partition functions,

$$Z_{NS,\mathcal{L}_{2,1}} = 2\chi_{2,1}\overline{\chi_{1,1}} + 2\chi_{2,1}\overline{\chi_{3,1}}, \quad Z_{NS,(-1)^F \mathcal{L}_{2,1}} = 2\chi_{1,1}\overline{\chi_{2,1}} + 2\chi_{3,1}\overline{\chi_{2,1}},$$
$$Z_{NS,(-1)^F} = 2\chi_{2,1}\overline{\chi_{2,1}}. \tag{5.12}$$

From the relations (5.11), we find the defect partition functions for $(-1)^{F_L}$ and $(-1)^{F_R}$,

$$Z_{NS,(-1)^{F_L}} = \sqrt{2}\chi_{2,1}\overline{\chi_{1,1}} + \sqrt{2}\chi_{2,1}\overline{\chi_{3,1}}, \quad Z_{NS,(-1)^{F_R}} = \sqrt{2}\chi_{1,1}\overline{\chi_{2,1}} + \sqrt{2}\chi_{3,1}\overline{\chi_{2,1}}, \tag{5.13}$$

where the degeneracies are properly quantized as expected.

The subset of TDLs $\{1, \mathcal{L}_{2,1}\}$ (as well as the one with $\{1, (-1)^F \mathcal{L}_{2,1}\}$) realizes the super fusion category $\mathcal{C}_q^0$. More precisely, by comparing the spin contents in (5.12) with the spin selection rule (4.20), we conclude that $\{1, \mathcal{L}_{2,1}\}$ realizes the solution (3.23) with $\kappa = 1, \gamma = e^{\frac{i3\pi}{4}}$ [24] and $\{1, (-1)^F \mathcal{L}_{2,1}\}$ realizes the solution (3.23) with $\kappa = 1, \gamma = e^{\frac{i\pi}{4}}$.

### 5.1.2 Tensor product of $\nu$ copies of $m = 3$ models

Let us consider the tensor product of $\nu$ copies of $m = 3$ models (free Majorana fermions). We focus on the TDLs

$$(-1)^{F_L} \equiv (-1)^{F_L^{(1)}} \cdots (-1)^{F_L^{(\nu)}}, \tag{5.15}$$

where $(-1)^{F_L^{(i)}}$ is the left fermion parity of the $i$-th copy. The defect partition function of $(-1)^{F_L}$ is simply given by the $\nu$-th power of the single-copy defect partition function,

$$Z_{NS,(-1)^{F_L}} = 2^{\frac{\nu}{2}} \left( \chi_{2,1}\overline{\chi_{1,1}} + \chi_{2,1}\overline{\chi_{3,1}} \right)^{\nu}. \tag{5.16}$$

From the above partition function, we read off the spin content of the defect Hilbert space $\mathcal{H}_{(-1)^{F_L}}$ as

$$s \in \frac{1}{2}\mathbb{Z} + \frac{\nu}{16}. \tag{5.17}$$

---

[24] As pointed out by [92], this $(-1)^{F_L}$ chiral $\mathbb{Z}_2$ symmetry can be implemented by the stacking the Kitaev chain with a fermionic CFT, whose partition function is given by the Arf invariant $\text{Arf}(\rho)$ associated with a spin structure $\rho$ (more precisely, a modified spin structure $S \cdot \rho$ twisted by a background $\mathbb{Z}_2$ gauge field $S$)

$$Z_{\text{INFO}} = \exp(i\pi \text{Arf}(\rho)). \tag{5.14}$$

Note that this stacking does not change the (torus) partition function.

By matching with the spin selection rules for $\mathcal{C}_{\mathrm{m}}^{0,0}$ (given in (4.12) of [8] with $n = 2$), $\widehat{\mathcal{C}}_{\mathrm{m}}^{0,0}$ (given in (4.17)) and $\mathcal{C}_{\mathrm{q}}^0$ (given in (4.20)), we see that $\{I, (-1)^{F_L}\}$ in the tensor product of $\nu$ copies of $m = 3$ models realizes all the rank-2 super fusion categories of $\mathbb{Z}_2$ symmetries (as summarized in Table 1).

### 5.1.3   $m = 4$

This model has four m-type and four q-type TDLs, whose fusion ring is generated by two m-type TDLs $\{(-1)^F, \mathcal{L}_{1,2}\}$ and a q-type TDL $\mathcal{L}_{2,1}$ with the relations[25]

$$((-1)^F)^2 = I, \quad \mathcal{L}_{2,1}^2 = (1_b + 1_f)I, \quad \mathcal{L}_{1,2}^2 = I + \mathcal{L}_{1,2}. \tag{5.19}$$

Taking the S-transformation of the twisted partition functions (5.4) and (5.5), we obtained the defect partition functions,

$$
\begin{aligned}
Z_{NS,\mathcal{L}_{2,1}} &= 2\chi_{2,1}\overline{\chi_{1,1}} + 2\chi_{2,2}\overline{\chi_{1,2}} + 2\chi_{2,1}\overline{\chi_{3,1}} + 2\chi_{2,2}\overline{\chi_{3,2}}, \\
Z_{NS,(-1)^F\mathcal{L}_{2,1}} &= 2\chi_{1,1}\overline{\chi_{2,1}} + 2\chi_{3,1}\overline{\chi_{2,1}} + 2\chi_{1,2}\overline{\chi_{2,2}} + 2\chi_{3,2}\overline{\chi_{2,2}}, \\
Z_{NS,\mathcal{L}_{1,2}} &= \chi_{1,2}\overline{\chi_{1,1}} + \chi_{3,2}\overline{\chi_{1,1}} + \chi_{1,1}\overline{\chi_{1,2}} + \chi_{1,2}\overline{\chi_{1,2}} + \chi_{3,1}\overline{\chi_{1,2}} + \chi_{3,2}\overline{\chi_{1,2}} \\
&\quad + \chi_{1,2}\overline{\chi_{3,1}} + \chi_{3,2}\overline{\chi_{3,1}} + \chi_{1,1}\overline{\chi_{3,2}} + \chi_{1,2}\overline{\chi_{3,2}} + \chi_{3,1}\overline{\chi_{3,2}} + \chi_{3,2}\overline{\chi_{3,2}},
\end{aligned}
\tag{5.20}
$$

The super fusion category $\mathcal{C}_{\mathrm{m}}^{(1,0)}$ of $\zeta = \frac{1+\sqrt{5}}{2}$ is realized by the TDLs $\{I, \mathcal{L}_{1,2}\}$ [16], and we find that the spin selection rules (4.8) and (4.9) are indeed satisfied by the defect Hilbert space $\mathcal{H}_{\mathcal{L}_{1,2}}$ given by the defect partition function $Z_{NS,\mathcal{L}_{1,2}}$. On the other hand, the super fusion categories $\mathcal{C}_{\mathrm{q}}^0$ with $\kappa = 1, \gamma = \pm e^{\pm\frac{i\pi}{4}}$ are realized by the subsets of TDLs $\{I, \mathcal{L}_{2,1}\}$ and $\{I, (-1)^F\mathcal{L}_{2,1}\}$.

### 5.1.4   $m = 7$

This model has eighteen m-type TDLs and six q-type TDLs. Let us focus on the subset of TDLs $\{I, \mathcal{L}_{3,1}\}$, which has the fusion rule

$$\mathcal{L}_{3,1}^2 = I + 2\mathcal{L}_{3,1}. \tag{5.21}$$

From the formula of the twisted partition function (5.4), we find the quantum dimension of $\mathcal{L}_{3,1}$ is

$$\langle \mathcal{L}_{3,1} \rangle_{\mathbb{R}^2} = \frac{S_{(3,1),(1,1)}}{S_{(1,1),(1,1)}} = 1 + \sqrt{2}. \tag{5.22}$$

---

[25]The translation between the notation in [16] and here is

$$R = \frac{1}{\sqrt{2}}\mathcal{L}_{2,1}, \quad W = \mathcal{L}_{1,2}, \tag{5.18}$$

where note that the TDL $R$ is a rescaled q-type TDL.

Hence, it realizes the super fusion category $\mathcal{C}_{\mathrm{m}}^{(1,1)}$ with $w = 1 + \sqrt{2}$. Indeed, the defect partition function of $\mathcal{L}_{3,1}$ is

$$
\begin{aligned}
Z_{NS,\mathcal{L}_{3,1}} =\, & \chi_{1,1}\overline{\chi_{3,1}} + \chi_{1,1}\overline{\chi_{5,1}} + \chi_{1,2}\overline{\chi_{3,2}} + \chi_{1,2}\overline{\chi_{5,2}} + \chi_{1,3}\overline{\chi_{3,3}} + \chi_{1,3}\overline{\chi_{5,3}} + \chi_{3,1}\overline{\chi_{1,1}} \\
& + 2\chi_{3,1}\overline{\chi_{3,1}} + 2\chi_{3,1}\overline{\chi_{5,1}} + \chi_{3,1}\overline{\chi_{7,1}} + \chi_{3,2}\overline{\chi_{1,2}} + 2\chi_{3,2}\overline{\chi_{3,2}} + 2\chi_{3,2}\overline{\chi_{5,2}} \\
& + \chi_{3,2}\overline{\chi_{7,2}} + \chi_{3,3}\overline{\chi_{1,3}} + 2\chi_{3,3}\overline{\chi_{3,3}} + 2\chi_{3,3}\overline{\chi_{5,3}} + \chi_{3,3}\overline{\chi_{7,3}} + \chi_{5,1}\overline{\chi_{1,1}} \\
& + 2\chi_{5,1}\overline{\chi_{3,1}} + 2\chi_{5,1}\overline{\chi_{5,1}} + \chi_{5,1}\overline{\chi_{7,1}} + \chi_{5,2}\overline{\chi_{1,2}} + 2\chi_{5,2}\overline{\chi_{3,2}} + 2\chi_{5,2}\overline{\chi_{5,2}} \\
& + \chi_{5,2}\overline{\chi_{7,2}} + \chi_{5,3}\overline{\chi_{1,3}} + 2\chi_{5,3}\overline{\chi_{3,3}} + 2\chi_{5,3}\overline{\chi_{5,3}} + \chi_{5,3}\overline{\chi_{7,3}} + \chi_{7,1}\overline{\chi_{3,1}} \\
& + \chi_{7,1}\overline{\chi_{5,1}} + \chi_{7,2}\overline{\chi_{3,2}} + \chi_{7,2}\overline{\chi_{5,2}} + \chi_{7,3}\overline{\chi_{3,3}} + \chi_{7,3}\overline{\chi_{5,3}}\,,
\end{aligned}
\tag{5.23}
$$

whose spin content agrees with the spin selection rule (4.15).

## 5.2 Exceptional models

There are two exceptional fermionic minimal models of $m = 11$ and $m = 12$ found in [79]. Their NS-sector partition functions are given by[26]

$$
Z_{NS} = \sum_{s=1}^{\frac{p-1}{2}} |\chi_{1,s} + \chi_{5,s} + \chi_{7,s} + \chi_{11,s}|^2 \,.
\tag{5.24}
$$

Both models realize the $\mathcal{C}_{\mathrm{q}}^2$ super fusion category, which has a q-type TDL $W$ with the fusion rule

$$
W^2 = (1_{\mathrm{b}} + 1_{\mathrm{f}})I + 2W \,.
\tag{5.25}
$$

The NS partition functions twisted by the q-type TDL $W$ are given by[27]

$$
\begin{aligned}
Z_{NS}^W =\, & \sum_{s=1}^{\frac{p-1}{2}} (\chi_{1,s} + \chi_{5,s} + \chi_{7,s} + \chi_{11,s})^* \\
& \times \left( (1 + \sqrt{3})(\chi_{1,s} + \chi_{11,s}) + (1 - \sqrt{3})(\chi_{5,s} + \chi_{7,s}) \right) \,.
\end{aligned}
\tag{5.26}
$$

The partition function of the defect Hilbert space $\mathcal{H}_W$ is given by the modular S transformation of $Z_{NS}^W$, i.e. $Z_{NS,W}(\tau, \bar{\tau}) = Z_{NS}^W(-\frac{1}{\tau}, -\frac{1}{\bar{\tau}})$. One can check that $Z_{NS,W}(\tau, \bar{\tau})$ indeed admits an integer expansion of $q = e^{2\pi i \tau}$ and $\bar{q}$. The spin content of $\mathcal{H}_W$ is

$$
s \in \mathbb{Z} +
\begin{cases}
0, \frac{1}{6}, \frac{1}{2}, \frac{2}{3} & \text{for } m = 11\,, \\
0, \frac{1}{3}, \frac{1}{2}, \frac{5}{6} & \text{for } m = 10\,.
\end{cases}
\tag{5.27}
$$

---

[26]The NS-sector partition functions for $m = 10$ and $m = 11$ models are given by the same formula.

[27]The partition functions twisted by $W$ for $m = 10$ and $m = 11$ models are given by the same formula.

The quantum dimension of $W$ is

$$\langle W \rangle_{\mathbb{R}^2} = 1 + \sqrt{3}. \tag{5.28}$$

Hence, by matching with the spin selection rule (4.26) and the quantum dimension (3.27), we find that the exceptional $m = 10$ and $m = 11$ realizes the $\mathcal{C}_q^2$ super fusion category with $\zeta = 1 - \sqrt{3}$ and $\gamma = \pm e^{\pm \frac{i\pi}{4}}$.

# 6  Summary and discussions

This paper is devoted to developing the theory of TDLs in fermionic CFTs.

1. We formulated the defining properties of TDLs in 2d fermionic CFTs, which are mostly carried over from those for 2d bosonic CFTs, except that the Hilbert spaces associated with junctions and endpoints of TDLs can now also host fermionic operators. In addition, there is a new type of simple TDLs (q-type TDLs), whose two-way junction vector spaces host fermionic operator of conformal weights $(h, \tilde{h}) = (0, 0)$.

2. We derived several consequences from the defining properties and discussed their relation to super fusion categories. In particular, the F-moves (crossings) (2.17) of the TDLs are constrained by the projection condition (2.18) and the super pentagon identity (2.20). The later differs from the pentagon identity in the bosonic CFT by an extra edge (the map $S_{2,1,i}^{4,3,k}$) in the commutative diagram (2.19), that exchanges the order of two junction vectors.

3. We gave a conjectural classification of the rank-2 super fusion categories. The nontrivial invertible categories are $\mathcal{C}_m^{(0,0)}$, $\widehat{\mathcal{C}}_m^{(0,0)}$, and $\mathcal{C}_q^0$. The nontrivial non-invertible categories are $\mathcal{C}_m^{(1,0)}$, $\mathcal{C}_m^{(0,1)}$, $\mathcal{C}_m^{(1,1)}$, and $\mathcal{C}_q^2$. We gave the F-moves explicitly for all of them.

4. We derived the spin selection rules for the defect Hilbert spaces of the TDLs in the aforementioned nontrivial super fusion categories, and discussed their realizations in the fermionic minimal models.

5. We found the full set of simple TDLs in the standard fermionic minimal models and a partial set of TDLs in the exceptional fermionic minimal models. These TDLs realize all the rank-2 super fusion categories with a positive loop expectation value except $\mathcal{C}_m^{(1,1)}$ (see Table 2).

Finally, we comment on the constraints from the TDLs on the RG flows between fermionic fixed points. A TDL is preserved along an RG flow if it commutes with the conformal primary

that triggers the RG flow. If a unitary RG flow preserves a TDL with non-integer quantum dimension, then the IR phase cannot be a non-degenerate gapped state [8]. Following the argument in [8], we derive a refinement of this statement for the preserving TDL being q-type. Let $\mathcal{L}_q$ be a q-type TDL preserved along an RG flow. Suppose the IR fixed point is a TQFT with a unique vacuum, it follows that

$$\langle \mathcal{L}_q \rangle_{S^1 \times \mathbb{R}} = \mathrm{Tr}\, \widehat{\mathcal{L}}_q = \mathrm{Tr}_{\mathcal{H}_{\mathcal{L}_q}} 1 = \dim \mathcal{H}_{\mathcal{L}_q}\,, \tag{6.1}$$

where on the second equality we use the modular S transformation. The defect Hilbert space $\mathcal{H}_{\mathcal{L}_q}$ is even dimensional. Hence, we find

$$\langle \mathcal{L}_q \rangle_{S^1 \times \mathbb{R}} \in 2\mathbb{Z}_{\geq 0}\,. \tag{6.2}$$

In other words, an RG flow cannot be trivially gapped if it preserves a q-type TDL whose quantum dimension is not a non-negative even integer.

# Acknowledgements

We would like to thank Ken Kikuchi for collaboration during the early stage of this project. We thank Yongchao Lü and Qing-Rui Wang for insightful discussions. CC is partly supported by National Key R&D Program of China (NO. 2020YFA0713000). The work of J.C. is supported by the National Thousand-Young-Talents Program of China. F. Xu is partly supported by the Research Fund for International Young Scientists, NSFC grant No. 11950410500 and the Young Scientists Fund of NSFC under the grant No. 12205159.

# A    General solution to the projection condition

In this appendix, we give a solution to the projection condition (2.18). As in (2.8), the pair-creation process on the internal line of the H-junctions in (2.17) gives relations

$$\tag{A.1}$$

where for simplicity we do not label the external and internal lines, which can be distinct TDLs in general. We have focused on the case where the internal lines are q-type, and the black dots at the junction denote the fermionic junction vectors. The $X, Y, Z, W$ are matrix representations of the $\Psi_{\overline{\mathcal{L}}} \otimes \Psi_{\mathcal{L}}$ in (2.8). The relations (A.1) can be rephrased as the projection conditions

$$
\left(\begin{array}{c} \end{array}\right) = P_5 \left(\begin{array}{c} \end{array}\right) , \qquad \left(\begin{array}{c} \end{array}\right) = P_6 \left(\begin{array}{c} \end{array}\right) , \tag{A.2}
$$

where the projection matrices $P_5$ and $P_6$ are

$$
P_5 = \frac{1}{2} \begin{pmatrix} 1 & X & 0 & 0 \\ X^{-1} & 1 & 0 & 0 \\ 0 & 0 & 1 & Y \\ 0 & 0 & Y^{-1} & 1 \end{pmatrix} , \quad P_6 = \frac{1}{2} \begin{pmatrix} 1 & Z & 0 & 0 \\ Z^{-1} & 1 & 0 & 0 \\ 0 & 0 & 1 & W \\ 0 & 0 & W^{-1} & 1 \end{pmatrix} . \tag{A.3}
$$

The solution to the projection condition (2.18) in the matrix representation is

$$
\left(\begin{array}{c} \end{array}\right) = \begin{pmatrix} 1 & 0 \\ X^{-1} & 0 \\ 0 & 1 \\ 0 & Y^{-1} \end{pmatrix} \begin{pmatrix} f_b & 0 \\ 0 & f_f \end{pmatrix} \begin{pmatrix} 1 & Z & 0 & 0 \\ 0 & 0 & 1 & W \end{pmatrix} \left(\begin{array}{c} \end{array}\right) , \tag{A.4}
$$

where for simplicity, we have assumed that the internal lines on both sides of the F-move are q-type. It is straightforward to generalize the above formula to the case when m-type internal lines appear on either side of the equations.

# B    Universal sector for oriented q-type TDLs

In this appendix, we summarize the solution for the universal sector of an oriented q-type TDL $\mathcal{L}$. With a suitable gauge choice, the 1d Majorana fermion on $\mathcal{L}$ acts on the junction

vector spaces $V_{I,\mathcal{L},\overline{\mathcal{L}}}$, $V_{I,\overline{\mathcal{L}},\mathcal{L}}$, $V_{\mathcal{L},I,\overline{\mathcal{L}}}$, $V_{\overline{\mathcal{L}},I,\mathcal{L}}$, $V_{\mathcal{L},\overline{\mathcal{L}},I}$, $V_{\overline{\mathcal{L}},\mathcal{L},I}$ as

$$\tag{B.1}$$

After solving the projection condition, the F-matrices are of the form

$$\mathcal{F}_I^{\mathcal{L},\overline{\mathcal{L}},I}(I,\overline{\mathcal{L}}) = \begin{pmatrix} f_{1b} & 0 \\ 0 & f_{1f} \end{pmatrix} \begin{pmatrix} 1 & -\frac{\gamma}{\tilde\alpha} & 0 & 0 \\ 0 & 0 & 1 & \gamma\tilde\alpha \end{pmatrix}, \quad \mathcal{F}_I^{\mathcal{L},I,\overline{\mathcal{L}}}(\mathcal{L},\overline{\mathcal{L}}) = \begin{pmatrix} 1 & 0 \\ \alpha\gamma & 0 \\ 0 & 1 \\ 0 & \frac{\gamma}{\alpha} \end{pmatrix} \begin{pmatrix} f_{2b} & 0 \\ 0 & f_{2f} \end{pmatrix} \begin{pmatrix} 1 & -\frac{\gamma}{\tilde\beta} & 0 & 0 \\ 0 & 0 & 1 & \tilde\beta\gamma \end{pmatrix},$$

$$\mathcal{F}_{\mathcal{L}}^{\mathcal{L},I,I}(\mathcal{L},I) = \begin{pmatrix} 1 & 0 \\ 1 & 0 \\ 0 & 1 \\ 0 & \frac{1}{\alpha^2} \end{pmatrix} \begin{pmatrix} f_{3b} & 0 \\ 0 & f_{3f} \end{pmatrix}, \quad \mathcal{F}_I^{I,\mathcal{L},\overline{\mathcal{L}}}(\mathcal{L},I) = \begin{pmatrix} 1 & 0 \\ \beta\gamma & 0 \\ 0 & 1 \\ 0 & \frac{\gamma}{\beta} \end{pmatrix} \begin{pmatrix} f_{4b} & 0 \\ 0 & f_{4f} \end{pmatrix},$$

$$\mathcal{F}_{\mathcal{L}}^{I,\mathcal{L},I}(\mathcal{L},\mathcal{L}) = \begin{pmatrix} 1 & 0 \\ \frac{\beta}{\alpha} & 0 \\ 0 & 1 \\ 0 & \frac{1}{\beta\alpha} \end{pmatrix} \begin{pmatrix} f_{5b} & 0 \\ 0 & f_{5f} \end{pmatrix} \begin{pmatrix} 1 & \frac{\beta}{\alpha} & 0 & 0 \\ 0 & 0 & 1 & \alpha\beta \end{pmatrix}, \quad \mathcal{F}_{\mathcal{L}}^{I,I,\mathcal{L}}(I,\mathcal{L}) = \begin{pmatrix} f_{6b} & 0 \\ 0 & f_{6f} \end{pmatrix} \begin{pmatrix} 1 & 1 & 0 & 0 \\ 0 & 0 & 1 & \beta^2 \end{pmatrix},$$

$$\tag{B.2}$$

and

$$\mathcal{F}_I^{\overline{\mathcal{L}},\mathcal{L},I}(I,\mathcal{L}) = \begin{pmatrix} \tilde{f}_{1b} & 0 \\ 0 & \tilde{f}_{1f} \end{pmatrix} \begin{pmatrix} 1 & \frac{\tilde{\gamma}}{\alpha} & 0 & 0 \\ 0 & 0 & 1 & \tilde{\gamma}\alpha \end{pmatrix}, \quad \mathcal{F}_I^{\overline{\mathcal{L}},I,\mathcal{L}}(\overline{\mathcal{L}},\mathcal{L}) = \begin{pmatrix} 1 & 0 \\ -\tilde{\alpha}\tilde{\gamma} & 0 \\ 0 & 1 \\ 0 & \frac{\tilde{\gamma}}{\tilde{\alpha}} \end{pmatrix} \begin{pmatrix} \tilde{f}_{2b} & 0 \\ 0 & \tilde{f}_{2f} \end{pmatrix} \begin{pmatrix} 1 & \frac{\tilde{\gamma}}{\tilde{\beta}} & 0 & 0 \\ 0 & 0 & 1 & \beta\tilde{\gamma} \end{pmatrix},$$

$$\mathcal{F}_{\overline{\mathcal{L}}}^{\overline{\mathcal{L}},I,I}(\overline{\mathcal{L}},I) = \begin{pmatrix} 1 & 0 \\ -1 & 0 \\ 0 & 1 \\ 0 & \frac{1}{\tilde{\alpha}^2} \end{pmatrix} \begin{pmatrix} \tilde{f}_{3b} & 0 \\ 0 & \tilde{f}_{3f} \end{pmatrix}, \quad \mathcal{F}_I^{I,\overline{\mathcal{L}},\mathcal{L}}(\overline{\mathcal{L}},I) = \begin{pmatrix} 1 & 0 \\ -\tilde{\beta}\tilde{\gamma} & 0 \\ 0 & 1 \\ 0 & \frac{\tilde{\gamma}}{\tilde{\beta}} \end{pmatrix} \begin{pmatrix} \tilde{f}_{4b} & 0 \\ 0 & \tilde{f}_{4f} \end{pmatrix},$$

$$\mathcal{F}_{\overline{\mathcal{L}}}^{I,\overline{\mathcal{L}},I}(\overline{\mathcal{L}},\overline{\mathcal{L}}) = \begin{pmatrix} 1 & 0 \\ -\frac{\tilde{\beta}}{\tilde{\alpha}} & 0 \\ 0 & 1 \\ 0 & \frac{1}{\tilde{\beta}\tilde{\alpha}} \end{pmatrix} \begin{pmatrix} \tilde{f}_{5b} & 0 \\ 0 & \tilde{f}_{5f} \end{pmatrix} \begin{pmatrix} 1 & -\frac{\tilde{\beta}}{\tilde{\alpha}} & 0 & 0 \\ 0 & 0 & 1 & \tilde{\alpha}\tilde{\beta} \end{pmatrix}, \quad \mathcal{F}_{\overline{\mathcal{L}}}^{I,I,\overline{\mathcal{L}}}(I,\overline{\mathcal{L}}) = \begin{pmatrix} \tilde{f}_{6b} & 0 \\ 0 & \tilde{f}_{6f} \end{pmatrix} \begin{pmatrix} 1 & -1 & 0 & 0 \\ 0 & 0 & 1 & \tilde{\beta}^2 \end{pmatrix}.$$

$$(B.3)$$

By solving the super pentagon equations, we find the following two gauge inequivalent solutions,

$$\alpha = \tilde{\alpha} = \beta = \tilde{\beta} = \tilde{\gamma} = 1, \quad f_{1b} = \tilde{f}_{1b} = \tilde{f}_{1f} = f_{2b} = \tilde{f}_{2b} = \tilde{f}_{2f} = \frac{1}{2},$$

$$f_{1f} = \frac{\gamma^3}{2}, \quad f_{2f} = \frac{\gamma^2}{2}, \quad f_{3b} = \tilde{f}_{3b} = f_{3f} = \tilde{f}_{3f} = f_{4b} = \tilde{f}_{4b} = \tilde{f}_{4f} = 1, \quad (B.4)$$

$$f_{4f} = \gamma^3, \quad f_{5b} = \tilde{f}_{5b} = f_{5f} = \tilde{f}_{5f} = f_{6b} = \tilde{f}_{6b} = f_{6f} = \tilde{f}_{6f} = \frac{1}{2},$$

for $\gamma = 1, i$.

# C  Fusion coefficients

In this appendix, we follow the appendix B of [8] to derive the relation between the fusion coefficients and the dimensions of the junction vector spaces summarized in the fusion rule

(2.42). First, consider the following manipulations of the TDL configurations

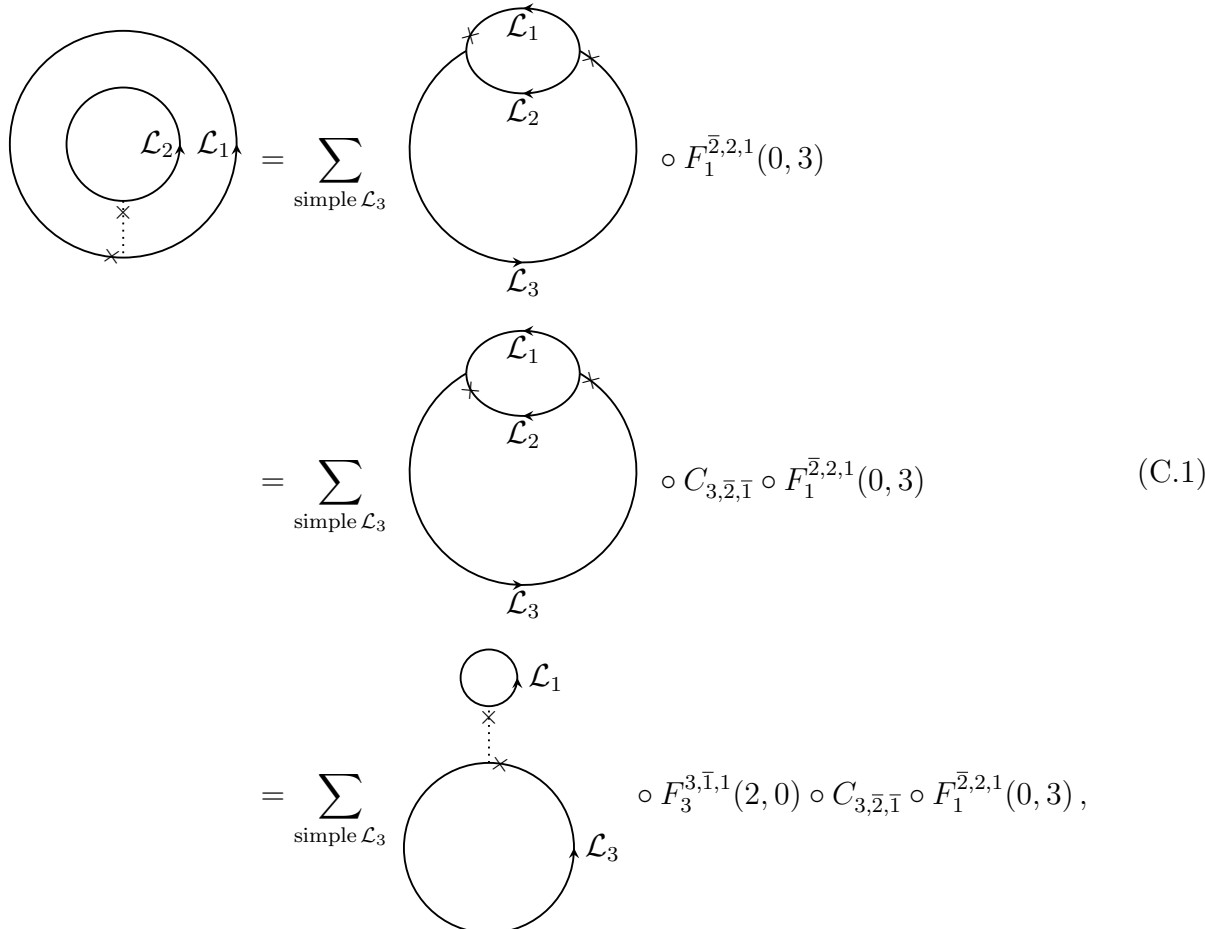

$$= \sum_{\text{simple } \mathcal{L}_3} \quad \circ C_{3,\bar{2},\bar{1}} \circ F_1^{\bar{2},2,1}(0,3) \qquad (C.1)$$

$$= \sum_{\text{simple } \mathcal{L}_3} \quad \circ F_3^{3,\bar{1},1}(2,0) \circ C_{3,\bar{2},\bar{1}} \circ F_1^{\bar{2},2,1}(0,3) \,,$$

where the 0's denote the trivial line. Therefore, we find that the identity of the fusion coefficient $N_{\mathcal{L}_1,\mathcal{L}_2}^{\mathcal{L}_3}$,

$$N_{\mathcal{L}_1,\mathcal{L}_2}^{\mathcal{L}_3} = \frac{1}{\langle \mathcal{L}_1 \rangle_{\mathbb{R}^2}} \langle 1_{1,\bar{1},0}, 1_{0,3,\bar{3}} | F_3^{3,\bar{1},1}(2,0) \circ C_{3,\bar{2},\bar{1}} \circ F_1^{\bar{2},2,1}(0,3) | 1_{2,\bar{2},0}, 1_{1,0,\bar{1}} \rangle \,, \qquad (C.2)$$

where sandwiching by the bra $\langle 1_{1,\bar{1},0}, 1_{0,3,\bar{3}} |$ and ket $| 1_{2,\bar{2},0}, 1_{1,0,\bar{1}} \rangle$ means evaluating on the corresponding identity junction vectors.

Next, consider the pentagon identity

$$F_3^{3,\bar{1},1}(2,0) \circ F_1^{\bar{2},2,1}(0,3) \circ F_0^{\bar{2},3,\bar{1}}(1,2) = F_1^{\bar{2},3,0}(1,3) \circ S_{3,\bar{2},1}^{1,\bar{1},0} \circ F_1^{1,\bar{1},1}(0,0) \,, \qquad (C.3)$$

which follows from the commutative diagram

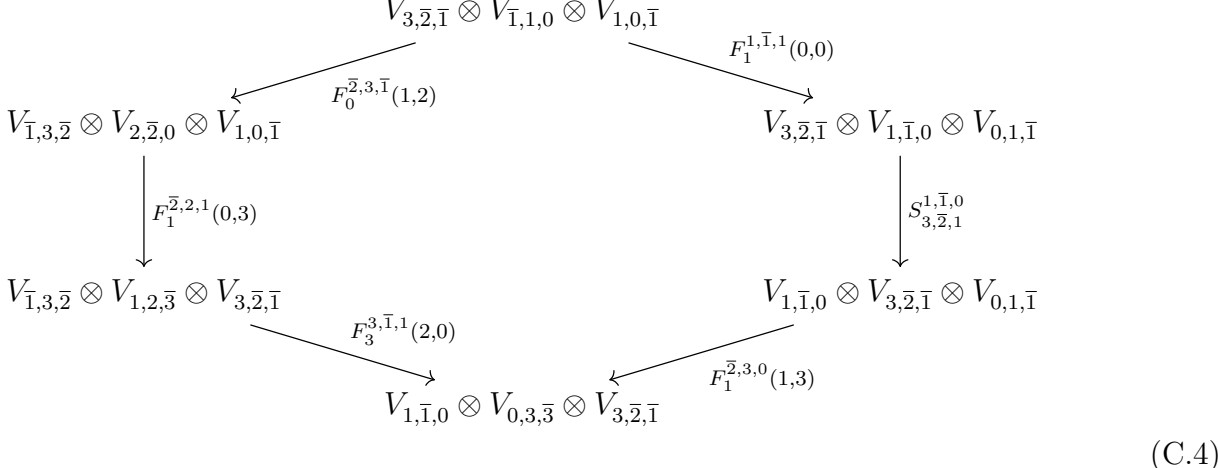

$$\text{(C.4)}$$

Let us follow the right route of this diagram starting from a junction vector $v \in V_{3,\bar{2},\bar{1}}$ and two identity junction vectors in $V_{\bar{1},1,0}$ and $V_{1,0,\bar{1}}$,

$$v \otimes 1 \otimes 1 \mapsto \langle \mathcal{L}_1 \rangle_{\mathbb{R}^2} (1 \otimes v \otimes 1) \mapsto \langle \mathcal{L}_1 \rangle_{\mathbb{R}^2} (1 \otimes 1 \otimes v)$$

$$\mapsto \begin{cases} \langle \mathcal{L}_1 \rangle_{\mathbb{R}^2} (1 \otimes 1 \otimes v) & \text{if } \mathcal{L}_3 \text{ is m-type,} \\ \frac{\langle \mathcal{L}_1 \rangle_{\mathbb{R}^2}}{2} (1 \otimes 1 \otimes v + 1 \otimes \Psi_{\bar{3}}(1) \otimes \Psi_3(v)) & \text{if } \mathcal{L}_3 \text{ is q-type,} \end{cases} \qquad \text{(C.5)}$$

where we have used the relation (2.19) in [8] and ignored the subscripts of the identity junction vectors. This implies the following relation

$$\big\langle 1_{1,\bar{1},0}, 1_{0,3,\bar{3}} \big| \text{Tr}_{V_{3,\bar{2},1}} (F_1^{\bar{2},3,0}(1,3) \circ S_{3,\bar{2},1}^{1,\bar{1},0} \circ F_1^{1,\bar{1},1}(0,0)) \big| 1_{\bar{1},1,0}, 1_{1,0,\bar{1}} \big\rangle$$

$$= \begin{cases} \langle \mathcal{L}_1 \rangle_{\mathbb{R}^2} \dim_{\mathbb{C}}(V_{3,\bar{2},1}) & \text{if } \mathcal{L}_3 \text{ is m-type,} \\ \frac{\langle \mathcal{L}_1 \rangle_{\mathbb{R}^2}}{2} \dim_{\mathbb{C}}(V_{3,\bar{2},1}) & \text{if } \mathcal{L}_3 \text{ is q-type.} \end{cases} \qquad \text{(C.6)}$$

Next, let us follow the first two steps on the left route of the diagram,

$$v \otimes 1 \otimes 1$$

$$\mapsto \begin{cases} C_{3,\bar{2},\bar{1}}(v) \otimes 1 \otimes 1 & \text{if } \mathcal{L}_2 \text{ is m-type} \\ \frac{1}{2}(C_{3,\bar{2},\bar{1}}(v) \otimes 1 + \Psi_{\bar{2}}(C_{3,\bar{2},\bar{1}}(v)) \otimes \Psi_2(1)) \otimes 1 & \text{if } \mathcal{L}_2 \text{ is q-type} \end{cases}$$

$$\mapsto \begin{cases} C_{3,\bar{2},\bar{1}}(v) \otimes F_1^{\bar{2},2,1}(0,3)(1 \otimes 1) & \text{if } \mathcal{L}_2 \text{ is m-type} \\ \frac{1}{2} \left[ C_{3,\bar{2},\bar{1}}(v) \otimes F_1^{\bar{2},2,1}(0,3)(1 \otimes 1) + \Psi_{\bar{2}}(C_{3,\bar{2},\bar{1}}(v)) \otimes F_1^{\bar{2},2,1}(0,3)(\Psi_2(1) \otimes 1) \right] & \text{if } \mathcal{L}_2 \text{ is q-type} \end{cases}$$

$$= P_2(C_{3,\bar{2},\bar{1}}(v) \otimes F_1^{\bar{2},2,1}(0,3)(1 \otimes 1)),$$

$$\text{(C.7)}$$

where we have used the relation (2.20) in [8] between the F-symbol and the cyclic permutation map, and the fact that the action of the 1d Majorana fermion along an external line of an

H-junction commutes with the F-move, i.e. $\Psi_1 \circ F_4^{1,2,3}(5,6) = F_4^{1,2,3}(5,6) \circ \Psi_1$. Composing with the third step on the left route, we find the following identity

$$
\begin{aligned}
&\langle 1_{1,\bar{1},0}, 1_{0,3,\bar{3}} | F_3^{3,\bar{1},1}(2,0) \circ \mathrm{Tr}_{V_{3,\bar{2},1}}(F_1^{\bar{2},2,1}(0,3) \circ F_0^{\bar{2},3,\bar{1}}(1,2)) | 1_{\bar{1},1,0}, 1_{1,0,\bar{1}} \rangle \\
&= \langle 1_{1,\bar{1},0}, 1_{0,3,\bar{3}} | F_3^{3,\bar{1},1}(2,0) \circ C_{3,\bar{2},\bar{1}} \circ F_1^{\bar{2},2,1}(0,3) | 1_{2,\bar{2},0}, 1_{1,0,\bar{1}} \rangle .
\end{aligned}
\tag{C.8}
$$

Putting everything (C.2), (C.6), (C.8) together, we find

$$
N_{\mathcal{L}_1,\mathcal{L}_2}^{\mathcal{L}_3} = \begin{cases} \dim_{\mathbb{C}}(V_{3,\bar{2},1}) & \text{if } \mathcal{L}_3 \text{ is m-type,} \\ \dim_{\mathbb{C}^{1|1}}(V_{3,\bar{2},1}) & \text{if } \mathcal{L}_3 \text{ is q-type,} \end{cases}
\tag{C.9}
$$

where we have used the relation $\frac{1}{2}\dim_{\mathbb{C}}(V_{3,\bar{2},1}) = \dim_{\mathbb{C}^{1|1}}(V_{3,\bar{2},1})$.

# D   Detailed data of the $\mathcal{C}_{\mathrm{q}}^2$ category

## D.1   1d Majorana fermion action

The 1d Majorana fermion on $W$ acts on the junction vector space $V_{W,W,W}$ as

$$\tag{D.1}$$

where $\rho$, $\sigma$, $\tau$ are $2 \times 2$ matrices.

**Constraints on $\sigma$, $\rho$ and $\tau$-matrices**   Similar to the discussion in sec. 2.2.3, there are also additional constraints on the entries of projection matrices $\sigma$, $\rho$ and $\tau$. Let's consider

the following graphs,

$$\text{(D.2)}$$

where we have used (2.5) and (D.1) for the unoriented q-type line. Therefore we have the constraints:

$$\tau^{-1}\rho\,\tau^{-1}\rho = \tau^{-1}\sigma\,\tau^{-1}\sigma = -\rho^{-1}\sigma\,\rho^{-1}\sigma = 1\,. \tag{D.3}$$

In the derivations (D.2), there are potential sign ambiguities. Let us argue that they can actually be fixed by the universal sector solution (2.33). Considering the subgraphs of the graphs in (2.30) that contain only the upper trivalent junction, we find the relation

$$= \quad \epsilon_1 \tag{D.4}$$

where $\epsilon_1 = 1$ has been found in the universal sector. Now, we can sew the above graphs with the graphs in the pair-creation process on the left leg of the $WWW$-junction

$$= \quad \epsilon_L \tag{D.5}$$

where $\epsilon_L$ is the sign ambiguity. We find the following relations

$$= \epsilon_1 \qquad = \epsilon_L \tag{D.6}$$

Therefore, for consistency, we must have

$$\epsilon_L = \epsilon_1 = 1\,. \tag{D.7}$$

Similarly, one can argue that the sign ambiguities of the pair-creations on the other two legs of the $WWW$-junction

$$\text{(figure)} \quad = \quad \epsilon_R \ \text{(figure)} \quad , \qquad \text{(figure)} \quad = \quad \epsilon_D \ \text{(figure)} \quad , \tag{D.8}$$

should be fixed as

$$\epsilon_R = \epsilon_D = 1 \,, \tag{D.9}$$

and thus no sign ambiguities anymore.

**Gauge fixings on $\sigma$, $\rho$ and $\tau$-matrices**  Now we utilize gauges to further constrain the projection matrices. Notice that the junction vector space $V_{W,W,W}$ admits the gauge transformations

$$\text{(figure)} \quad \to \quad g_b \ \text{(figure)} \quad , \qquad \text{(figure)} \quad \to \quad g_f \ \text{(figure)} \quad , \tag{D.10}$$

where $g_b$ and $g_f$ are two $2 \times 2$ matrices. Combining with eq. (D.1), we see that the $\sigma$, $\rho$ and $\tau$-matrices transform accordingly as,

$$\sigma \longrightarrow g_b^{-1} \sigma \, g_f \,, \quad \rho \longrightarrow g_b^{-1} \rho \, g_f \,, \quad \text{and} \ \ \tau \longrightarrow g_b^{-1} \tau \, g_f \,. \tag{D.11}$$

First, one can use the gauges $g_b$ and $g_f$ to fix

$$\sigma = 1 \,, \tag{D.12}$$

and thus the constraints (D.3) turn out to be

$$\rho^2 = -1 \,, \quad \tau^2 = 1 \quad \text{and} \ \ \tau \rho = -\rho \tau \,. \tag{D.13}$$

In addition, there is a residual gauge left under the fixing of $\sigma = 1$, say letting $g_b = g_f \equiv R$, under which $\rho$ and $\tau$ further transform as

$$\rho \longrightarrow R^{-1} \rho R \,, \quad \text{and} \ \ \tau \longrightarrow R^{-1} \tau R \,. \tag{D.14}$$

Using it, we can diagonalize $\tau$-matrix, and solve the constraint (D.13) as

$$\sigma = \begin{pmatrix} 1 & 0 \\ 0 & 1 \end{pmatrix} , \quad \rho = \begin{pmatrix} 0 & \varrho \\ -\varrho^{-1} & 0 \end{pmatrix} , \quad \text{and} \ \ \tau = \pm \begin{pmatrix} 1 & 0 \\ 0 & -1 \end{pmatrix} , \tag{D.15}$$

where $\varrho$ is an undetermined $\mathbb{C}$-number. Actually, we did not use up all gauge freedoms in the residual gauge $R$. To see it, let us consider a further gauge transformation on eq. (D.15) by the $R$-gauge

$$R = \begin{pmatrix} R_{11} & R_{12} \\ R_{21} & R_{22} \end{pmatrix} , \tag{D.16}$$

while keeping the $\tau$-matrix invariant. One then finds that

$$R_{12} = R_{21} = 0\,, \quad \text{and} \quad \varrho \longrightarrow \frac{R_{22}}{R_{11}}\varrho\,. \tag{D.17}$$

We can use the ratio of $R_{11}$ and $R_{22}$ to rescale $\varrho = 1$. At last, the two solutions of the $\tau$-matrix are gauge equivalent as one can, after fixing $\varrho = 1$, assign a subsequent gauge transformation

$$R = \begin{pmatrix} 0 & 1 \\ -1 & 0 \end{pmatrix} \tag{D.18}$$

to interpolate $\tau$ to $-\tau$. Therefore we finally have

$$\sigma = \begin{pmatrix} 1 & 0 \\ 0 & 1 \end{pmatrix}, \quad \rho = \begin{pmatrix} 0 & 1 \\ -1 & 0 \end{pmatrix}, \quad \text{and} \quad \tau = \begin{pmatrix} 1 & 0 \\ 0 & -1 \end{pmatrix}. \tag{D.19}$$

## D.2 Solution to the projection condition

Let us first focus on the H-junctions with a single trivial external TDL $I$. When $I$ is the first external TDL, we have the projection conditions

$$\tag{D.20}$$

Solving these projection conditions, we find that the F-moves should take the form

$$\tag{D.21}$$

where the 1's are $2 \times 2$ identity matrices, and $f_{7b}$, $f_{7f}$ are $2 \times 2$ matrices. One can also write down the projection conditions in the case when $I$ being the other external lines, and solving them gives the following form of the F-moves

$$
\left( \vcenter{\hbox{}} \right) = \begin{pmatrix} 1 & 0 \\ \alpha\rho^{-1} & 0 \\ 0 & 1 \\ 0 & \alpha^{-1}\rho^{-1} \end{pmatrix} \begin{pmatrix} f_{8b} & 0 \\ 0 & f_{8f} \end{pmatrix} \begin{pmatrix} 1 & \beta^{-1}\sigma & 0 & 0 \\ 0 & 0 & 1 & \beta\sigma \end{pmatrix} \left( \vcenter{\hbox{}} \right) ,
$$

$$
\left( \vcenter{\hbox{}} \right) = \begin{pmatrix} 1 & 0 \\ \alpha^{-1}\tau^{-1} & 0 \\ 0 & 1 \\ 0 & \alpha^{-1}\tau \end{pmatrix} \begin{pmatrix} f_{9b} & 0 \\ 0 & f_{9f} \end{pmatrix} \begin{pmatrix} 1 & \alpha^{-1}\sigma & 0 & 0 \\ 0 & 0 & 1 & \alpha\sigma \end{pmatrix} \left( \vcenter{\hbox{}} \right) , \tag{D.22}
$$

$$
\left( \vcenter{\hbox{}} \right) = \begin{pmatrix} 1 & 0 \\ \gamma\tau^{-1} & 0 \\ 0 & 1 \\ 0 & \gamma\tau \end{pmatrix} \begin{pmatrix} f_{10b} & 0 \\ 0 & f_{10f} \end{pmatrix} \begin{pmatrix} 1 & \gamma\tau & 0 & 0 \\ 0 & 0 & 1 & \gamma\tau^{-1} \end{pmatrix} \left( \vcenter{\hbox{}} \right) .
$$

Finally, we consider the H-junctions with all the external TDLs are $W$. They satisfy the projection conditions

$$
\begin{aligned}
&\Yvertical = {}_1^2\Ydouble = (\tau \otimes \rho) \, {}^1_2\Ydouble \, , \\
&\Ydot = (\tau^{-1} \otimes 1) \Ydot = (\tau^{-1} \otimes \rho) \Ydot \, , \\
&\Ytri = \Ytri\,{}^1_2 = (\tau \otimes \sigma) \Ytri\,{}^1_2 \, , \\
&\Ytridot = (\tau^{-1} \otimes 1) \Ytridot = (\tau^{-1} \otimes \sigma) \Ytridot \, .
\end{aligned}
\tag{D.23}
$$

Solving these projection conditions, we find the following form for the F-moves

$$
\left(\text{diagram}\right) = \begin{pmatrix} 1\otimes 1 & 0 & 0 & 0 \\ \tau^{-1}\otimes\rho^{-1} & 0 & 0 & 0 \\ 0 & 1 & 0 & 0 \\ 0 & 0 & 1\otimes 1 & 0 \\ 0 & 0 & \tau\otimes\rho^{-1} & 0 \\ 0 & 0 & 0 & 1 \end{pmatrix} \begin{pmatrix} \mathcal{F}_b & 0 \\ 0 & \mathcal{F}_f \end{pmatrix}
$$

$$
\times \begin{pmatrix} 1\otimes 1 & \tau\otimes\sigma & 0 & 0 & 0 & 0 \\ 0 & 0 & 1 & 0 & 0 & 0 \\ 0 & 0 & 0 & 1\otimes 1 & \tau^{-1}\otimes\sigma & 0 \\ 0 & 0 & 0 & 0 & 0 & 1 \end{pmatrix} \left(\text{diagram}\right),
$$

(D.24)

where the 1's stand for the $2\times 2$ identity matrix and $\mathcal{F}_b$, $\mathcal{F}_f$ are $6\times 6$ matrices.

**Gauge fixings on $\mathcal{F}_f$**  By far one may notice that there is one more gauge freedom left, say the product of $R_{11}$ and $R_{22}$. Scanning the gauge transformation of the entries of various

F-symbols, see eq. (3.8), we pick a specific entry of F-symbols, say $\mathcal{F}_{W;0,1}^{W,W,W;1,0}(W,I)_{1,1}^{2,1}$. Its gauge transformation under $R$-matrix is given by

$$\mathcal{F}_{W;0,1}^{W,W,W;1,0}(W,I)_{1,1}^{2,1} \longrightarrow \frac{1}{R_{11}R_{22}}\mathcal{F}_{W;0,1}^{W,W,W;1,0}(W,I)_{1,1}^{2,1}. \tag{D.25}$$

Therefore, we fix the last gauge freedom by requiring

$$\mathcal{F}_{W;0,1}^{W,W,W;1,0}(W,I)_{1,1}^{2,1} = 1. \tag{D.26}$$

In terms of eq. (D.24), we have

$$\mathcal{F}_{f\,36} = \mathcal{F}_{W;0,1}^{W,W,W;1,0}(W,I)_{1,1}^{2,1} = 1. \tag{D.27}$$

# E  Super fusion category for $m = 4$ fermionic minimal model

In this subsection, we solve the super-pentagon equations for the super fusion category of $m = 4$ fermionic minimal model. As discussed in [16], we focus on the super fusion category generated by the lines $\widehat{W}$ and $R$,

$$\mathcal{C}_{m=4}^f = \left\langle \widehat{W}, R \right\rangle = \left\{ I, \widehat{W}, R, \widehat{W}R \right\}, \tag{E.1}$$

with the fusion rules

$$\widehat{W}^2 = I + \widehat{W}, \quad \text{and} \quad R^2 = I, \tag{E.2}$$

where $R$ and $\widehat{W}R$ are q-type TDLs. One can also establish this fermionic category by lifting the bosonic $m = 4$ minimal CFT to the 3d anyon theory and perform fermionic anyon condensation. In this context, the bosonic anyon theory has six Verlinde lines $\{\mathcal{L}_I, \mathcal{L}_\epsilon, \mathcal{L}_{\epsilon'}, \mathcal{L}_{\epsilon''}, \mathcal{L}_\sigma, \mathcal{L}_{\sigma'}\}$ corresponding to the six primaries $\{I, \epsilon, \epsilon', \epsilon'', \sigma, \sigma'\}$, where the line $\mathcal{L}_{\epsilon''}$ is a transparent fermionic anyon. After condensing $\mathcal{L}_{\epsilon''}$, $\mathcal{L}_I$ and $\mathcal{L}_\epsilon$ are identifed to $\mathcal{L}_{\epsilon''}$ and $\mathcal{L}_{\epsilon'}$ respectively, meanwhile $\mathcal{L}_\sigma$ and $\mathcal{L}_{\sigma'}$ go to themselves, and thus of q-type. We end up with the super fusion category consisted of four equivalent classes

$$\mathcal{C}_{m=4}^f = \{[\mathcal{L}_I], [\mathcal{L}_\epsilon], [\mathcal{L}_\sigma], [\mathcal{L}_{\sigma'}]\}. \tag{E.3}$$

One can check that the fusion algebra follows

$$[\mathcal{L}_\epsilon]^2 = [\mathcal{L}_I] + [\mathcal{L}_\epsilon], \quad [\mathcal{L}_\sigma]^2 = [\mathcal{L}_I], \quad \text{and} \quad [\mathcal{L}_\epsilon][\mathcal{L}_\sigma] = [\mathcal{L}_{\sigma'}], \tag{E.4}$$

i.e. eq. (E.2).

The super fusion category of fermionic $m = 4$ can be regarded as "tensor product" of two (super) fusion sub-categories of $\mathcal{C}_{\text{m}}^{(1,0)}$ and $\mathcal{C}_{\text{q}}^0$, as discussed in sec. 3 admit two and four solutions respectively. For the super pentagon equations of fermionic $m = 4$, there are 615 $F$-move entries constrained by 12850 equations. Among them, we can fix 31 gauge freedoms and finally find exactly eight gauge inequivalent solutions. Since the $F$-moves of lines $\widehat{W}$ and $R$ have been computed in sec. 3. We here only spell out the $F$-matrix of $\mathcal{L}_{\sigma'} \equiv \widehat{W}R$,

$$
\left( \begin{array}{c} \includegraphics \end{array} \right) = \sum_{b \in \{0,1\}} \mathcal{F}_{\mathcal{L}_{\sigma'}}^{\mathcal{L}_{\sigma'}\mathcal{L}_{\sigma'}\mathcal{L}_{\sigma'}} \left( \mathcal{L}_a, \mathcal{L}_b \right) \cdot \left( \begin{array}{c} \includegraphics \end{array} \right),
$$

$$
\mathcal{F}_{\mathcal{L}_{\sigma'}}^{\mathcal{L}_{\sigma'}\mathcal{L}_{\sigma'}\mathcal{L}_{\sigma'}} \left( \mathcal{L}_a, \mathcal{L}_b \right) = \frac{\kappa}{\sqrt{2}} \zeta(a,b) \cdot (-1)^{ab} \begin{pmatrix} 1 & 0 & 0 & -i\lambda \\ 0 & 1 & -i\lambda & 0 \\ 0 & 1 & i\lambda & 0 \\ 1 & 0 & 0 & i\lambda \end{pmatrix},
$$

$$
\text{or} \quad \mathcal{F}_{\mathcal{L}_{\sigma'}}^{\mathcal{L}_{\sigma'}\mathcal{L}_{\sigma'}\mathcal{L}_{\sigma'}} \left( \mathcal{L}_a, \mathcal{L}_b \right) = \frac{\kappa}{\sqrt{2}} \zeta^{-1}(a,b) \cdot (-1)^b \begin{pmatrix} 1 & 0 & 0 & -i\lambda \\ 0 & 1 & -i\lambda & 0 \\ 0 & 1 & i\lambda & 0 \\ 1 & 0 & 0 & i\lambda \end{pmatrix}, \tag{E.5}
$$

where the blue line denotes $\mathcal{L}_{\sigma'}$, the red lines stand for $\mathcal{L}_0 \equiv I$ and $\mathcal{L}_1 \equiv \widehat{W}$, and

$$
\zeta(a,b) \equiv \begin{cases} \frac{\sqrt{5}+1}{2}, & \text{for } (a,b) = (0,0),\, (1,0),\, (1,1) \\ 1, & \text{for } (a,b) = (0,1) \end{cases}. \tag{E.6}
$$

In the $F$-matrix, $\kappa = \pm 1$ is the Frobenius-Schur indicator of the line $\mathcal{L}_{\sigma'}$ and $\lambda = \pm 1$ corresponding to the two solutions in the universal sector of the line $R$. Overall there are eight solutions.

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
