# Peer review of "Topological Defect Lines in Two Dimensional Fermionic CFTs"

_SciPost Physics_

## Round 4 · Referee Report · Anonymous (Referee 1) · 2023-9-15

Report

The paper generalizes the concept of categorical symmetries, which are characterized by the presence of non-invertible topological defect lines (TDLs), to theories with fermions. Here are some of the new properties of TDLs in such theories that they discuss:

1) Because the lines can now end on fermionic operators, the usual pentagon identity is replaced by a super pentagon identity.

2) There is a new type of TDLs, called q-type TDLs, which does not exist in bosonic theories. One defining feature of this is that the identity term in the fusion rule comes with an extra factor of 2.

They give several examples of fermionic minimal models where such TDLs are present. They also present an application to constraining RG flows that preserve the TDL. Overall, I recommend this paper for publication because it introduces a novel framework for dealing with non-invertible symmetries in the presence of fermions. Even though the paper deals with theories in two dimensions, I expect some of the features to persist in higher dimensions.

---

## Round 4 · Referee Report · Anonymous (Referee 2) · 2023-10-6

Report

The manuscript investigates non-invertible topological lines in 1+1d fermionic CFTs. The manuscript discovers new structures compared to the topological lines in bosonic theories, such as decoration by Majorana zero mode. I recommend publication if the authors address the following mostly minor comments:

  • In well-defined non-chiral fermionic theories one can gauge the fermion parity symmetry to obtain bosonic theories, and the topological lines will become the topological lines previously studied. This method is extensively used to study fermionic SPT phases. Can the author comment on such approach to understanding topological lines in fermionic theories?

  • The manuscript discussed changing the normalization of the topological line operators. However, the fusion coefficients of topological operators are quantized (described by well-defined lower dimensional TQFTs), and such change of normalization does not seem compatible with the topological structure. Can the author clarify this?

  • validity: -
  • significance: -
  • originality: -
  • clarity: -
  • formatting: -
  • grammar: -

Author:  Jin Chen  on 2023-10-18  [id 4044]

(in reply to Report 2 on 2023-10-06)
Category:
answer to question

We thank the referee's comments. Followed are our repies. More detailed updates can be found in the list of changes.

For the first question, we add a paragraph in the end of summary and discussion section, which explains the observed relations between TDLs in bosonic and fermionic CFTs, as well as connections to the fermionic anyon condensation widely used in the condense matter community.

For the second question, the change of the normalization in (2.50) is only a formal manipulation to make the fusion rules look like a group multiplication. However, the fusion coefficients of the new fusion rules do not reflect the relation to the lower dimensional TQFT anymore, as the referee pointed out. We emphasize this by adding the word ``formal" before (2.50).

We also add a sentence below eq.(2.47) to explain the "factor of 2" in the fusion rule can be interpreted as an 1d TFT gapped from two 1d Majorana fermions.

---

## Editorial Decision

resubmitted